# Neuronal mTORC1 inhibition promotes longevity without suppressing anabolic growth and reproduction in *C. elegans*

**Hannah J. Smith**[1], **Anne Lanjuin**[1], **Arpit Sharma**[1], **Aditi Prabhakar**[1], **Ewelina Nowak**[1], **Peter G. Stine**[1], **Rohan Sehgal**[1], **Klement Stojanovski**[2], **Benjamin D. Towbin**[2], **William B. Mair**[1] *

**1** Dept. Molecular Metabolism, Harvard TH Chan School of Public Health, Massachusetts, United States of America, **2** Institute for Cell Biology, University of Bern, Bern, Switzerland

* wmair@hsph.harvard.edu

**Data Availability Statement:** RNA Sequencing data was been uploaded to GEO and can be accessed at: GSE237060.

## Abstract

mTORC1 (mechanistic target of rapamycin complex 1) is a metabolic sensor that promotes growth when nutrients are abundant. Ubiquitous inhibition of mTORC1 extends lifespan in multiple organisms but also disrupts several anabolic processes resulting in stunted growth, slowed development, reduced fertility, and disrupted metabolism. However, it is unclear if these pleiotropic effects of mTORC1 inhibition can be uncoupled from longevity. Here, we utilize the auxin-inducible degradation (AID) system to restrict mTORC1 inhibition to *C. elegans* neurons. We find that neuron-specific degradation of RAGA-1, an upstream activator of mTORC1, or LET-363, the ortholog of mammalian mTOR, is sufficient to extend lifespan in *C. elegans*. Unlike *raga-1* loss of function genetic mutations or somatic AID of RAGA-1, neuronal AID of RAGA-1 robustly extends lifespan without impairing body size, developmental rate, brood size, or neuronal function. Moreover, while degradation of RAGA-1 in all somatic tissues alters the expression of thousands of genes, demonstrating the widespread effects of mTORC1 inhibition, degradation of RAGA-1 in neurons only results in around 200 differentially expressed genes with a specific enrichment in metabolism and stress response. Notably, our work demonstrates that targeting mTORC1 specifically in the nervous system in *C. elegans* uncouples longevity from growth and reproductive impairments, and that many canonical effects of low mTORC1 activity are not required to promote healthy aging. These data challenge previously held ideas about the mechanisms of mTORC1 lifespan extension and underscore the potential of promoting longevity by neuron-specific mTORC1 modulation.

## Author summary

Inhibition of the metabolic pathway mTORC1 has been shown to extend lifespan in many organisms and has gained traction as a possible anti-aging therapeutic. However, in addition to lifespan extension, chronic and body-wide mTORC1 inhibition results in a downregulation of anabolic processes that leads to unwanted side effects such as impaired growth, development, and reproduction in *C. elegans*, *Drosophila melanogaster*, and mice.

**Funding:** This work was funded by F31AG076296 (HJS), R01AG059595 (WBM), R01AG067106 (WBM), and R01AG044346 (WBM). The funders had no role in study design, data collection and analysis, decision to publish, or preparation of the manuscript.

**Competing interests:** The authors have declared that no competing interests exist.

An open question in the field is if mTORC1 inhibition can be restricted to specific tissues to promote longevity while minimizing the anabolic trade-offs. In this study, we test this hypothesis using the model organism *C. elegans* and the auxin-inducible degradation (AID) system which provides many advantages over previous techniques used to study the tissue-specificity of mTORC1 in aging. We find that degradation of mTORC1 pathway components specifically in the neurons robustly extends lifespan while preserving normal growth, development, and reproduction. Furthermore, we find that animals with neuronal mTORC1 inhibition maintain functional sensory neurons and have changes in the expression of genes related to stress response and metabolism. This work challenges previously held theories that anabolic trade-offs are required for mTORC1 longevity and highlights a critical role for neuronal mTORC1 in the regulation of organismal aging.

## Introduction

As humans age they become susceptible to an increasing number of diseases that reduce quality of life. It is now widely accepted that the rate of biological aging is not fixed, but can be modulated via a variety of environmental, nutritional, genetic, and pharmacological interventions. Decades of research have converged on nutrient metabolism as a central regulator of aging [1]. A key metabolic pathway frequently implicated in organismal aging is the mechanistic target of rapamycin complex 1 (mTORC1) signaling pathway [2,3]. mTORC1 is a multiprotein complex that senses energy and nutrient availability to promote growth when resources are abundant. mTORC1 incorporates information about the availability of insulin, growth factors, amino acids, energy, and levels of cellular stress through upstream proteins like the Rag GTPases and the tuberous sclerosis complex (TSC) [4]. Once activated, mTORC1 promotes anabolic processes such as nucleotide and lipid synthesis while simultaneously inhibiting catabolic processes such as autophagy to drive cellular growth and development [4]. Inhibition of the mTORC1 pathway, either genetically or pharmacologically, has been shown to extend lifespan in yeast [5], worms [6,7], flies [8], and mice [9–11].

The success of mTORC1 inhibition in promoting the longevity of lab animals has made it a promising strategy for anti-aging interventions in humans, however, global mTORC1 inhibition also results in several negative pleiotropic effects. In invertebrates, genetic mutants with reduced mTORC1 signaling are long-lived but often smaller, develop slower, and are less fertile [11–13]. In mice, administration of the mTORC1 inhibitor rapamycin extends lifespan [9,14–16] but has also been documented to induce cataracts [14], male infertility [17], insulin resistance [18], and glucose intolerance [16]. In humans, administration of mTORC1 inhibitors results in decreased wound healing, reduced fertility, and, very commonly, metabolic disruptions like hyperglycemia and dyslipidemia [19]. Altogether, the off-target effects of global mTORC1 inhibition make it an unlikely clinical strategy to promote healthy aging.

Multiple theories suggest that suppression of anabolic phenotypes such as growth and reproduction might be causal to the longevity associated with mTORC1 inhibition. These theories fit the Y allocation model in which an organism has a fixed amount of resources to utilize, and shifting resources towards one process, such as somatic maintenance and longer lifespan, comes at the expense of another process, such as reproduction. These Y allocation theories propose that in times of resource scarcity, metabolic pathways that regulate energy allocation, such as mTORC1, are downregulated and may shunt more energy into self-preservation at the expense of growth and fertility so that organisms can survive until conditions are more favorable to reproduce [20,21]. However, the hypothesis that these trade-offs are causal to mTORC1

longevity has not been directly tested. Therefore, investigating whether disruptions to anabolic metabolism can be uncoupled from the anti-aging benefits of mTORC1 inhibition is a key step if modulation of this pathway is to be utilized to promote healthy aging in humans.

One strategy to uncouple mTORC1 longevity from anabolic trade-offs might be to identify the key tissue(s) in which mTORC1 acts to regulate lifespan. In *Drosophila melanogaster*, overexpression of the mTORC1 inhibitor *dTsc2* or overexpression of the dominant negative forms of *dTOR* (TOR) or *dS6K* (S6K1) specifically in the fat tissue extended lifespan [8]. Intestine-specific knockdown of the mTORC1 activator RagC using RNA interference against *ragc-1* extends lifespan in *C. elegans* [7]. Knockdown of *ifg-1* (the *C. elegans* ortholog of mammalian eIF-4G, a downstream target of mTORC1) specifically in the germline, neurons, or hypodermis is sufficient to extend lifespan. However, these tissue-specific knockdowns result in impaired fecundity and delayed growth, with germline *ifg-1* knockdown resulting in a total failure to reproduce [22].

Previous work from our lab has begun to address whether a tissue-specific mTORC1 intervention could promote longevity while minimizing trade-offs using a *C. elegans raga-1* loss-of-function mutant, a genetic model of mTORC1 inhibition. *raga-1* encodes the ortholog of mammalian RagA, a conserved upstream regulator of mTORC1 that activates the complex during amino acid abundance. The *raga-1* mutant is long-lived but also smaller, developmentally delayed, and less fertile, supporting the Y allocation theory. Rescuing *raga-1* expression specifically in the neurons of the otherwise *raga-1* loss-of-function animals suppressed their longevity, returning the animals to a wild type lifespan. In contrast, rescuing *raga-1* expression in the intestine did not alter the longevity phenotype. Furthermore, neuronal expression of *rsks-1* (ortholog of mammalian S6K1, a downstream target of mTORC1) also suppressed the longevity of a *rsks-1* loss-of-function mutant. Strikingly, the neuronal expression of *raga-1* did not rescue the stunted body size or slowed development phenotypes of the *raga-1* loss-of-function mutant [13]. Together, these findings suggest that the activation state of mTORC1 in the neurons is a key modulator of organismal longevity that is separable from regulation of growth defects. However, a key unknown is whether neuron-specific mTORC1 inhibition alone is sufficient to increase lifespan, and, if so, whether it can do so independently of the pleiotropic effects of global mTORC1 inhibition.

To test the hypothesis that targeting mTORC1 inhibition specifically in the neurons might uncouple longevity from the suppression of anabolic growth and development, we used the auxin-inducible degradation (AID) to achieve neuron-specific mTORC1 inhibition in *C. elegans* [23]. Remarkably, we found that neuronal degradation of RAGA-1 or LET-363/mTOR is sufficient to extend lifespan without impairing growth. We observed that degradation of RAGA-1 in all somatic tissues impairs development and reproduction as expected, but animals with neuronal degradation of RAGA-1 develop and reproduce the same as wild type animals. Moreover, selective degradation of RAGA-1 only in neurons does not induce the widespread transcriptional response seen upon somatic RAGA-1 degradation, but instead results in specific expression changes in genes involved in stress response and metabolism which are therefore coupled to lifespan extension.

Overall, this study reveals that mTORC1 regulates aging in *C. elegans* through neurons and does not require the suppression of growth or fertility to extend lifespan. Moreover, our tissue-specific model of mTORC1 inhibition is a valuable tool to identify the mechanisms that are causal to the resulting longevity.

## Results

### Generation of strains for somatic and neuronal mTORC1 inhibition

To test the hypothesis that mTORC1 regulates aging through its activity in the nervous system [13], we used the auxin-inducible degradation (AID) system to selectively degrade mTORC1

components in neurons, while leaving other sites of expression intact. The AID system, originally discovered in plants and adapted for protein degradation in *C. elegans*, offers multiple advantages over previous methods used to inhibit mTORC1: it allows for inducible, tissue-specific, and reversible degradation at the protein level. Briefly, TIR1 is a plant enzyme that interacts with endogenous *C. elegans* proteins to form an E3 ubiquitin ligase. In the presence of the plant hormone auxin, TIR1 binds AID-tagged proteins and ubiquitinates them, targeting them for degradation by the proteasome [23] (Fig 1A).

Using CRISPR/Cas9 gene editing, we tagged the 3' end of endogenous *raga-1* with the 135 base pair AID degron tag fused to codon-optimized EmGFP [23] allowing for both visualization and targeted degradation of RAGA-1::AID (Fig 1B–1D). RAGA-1::AID::EmGFP is abundant across all tissues in *C. elegans*, often in the form of small puncta (Fig 1C). Additionally, we tagged the 5' end of endogenous *let-363* with the AID sequence, referred to as LET-363:: AID (Fig 1B).

The tissue in which RAGA-1::AID and LET-363::AID will be degraded is determined by the expression of TIR1. We generated a neuron-specific TIR1 strain using the CRISPR-based SKILODGE (single copy knock-in loci for defined gene expression) system [24]. We knocked TIR1 into a SKILODGE cassette containing the pan-neuronal *rab-3* promoter and the *rab-3* 3' UTR to drive expression of TIR1 specifically in neurons, referred to as 'Neuronal::TIR1' (Fig 1B). We then crossed Neuronal::TIR1 into the RAGA-1::AID and LET-363::AID alleles to generate strains in which RAGA-1 or LET-363 can be degraded specifically in neurons upon exposure to auxin (Fig 1B). To compare the effects of loss of RAGA-1 or LET-363 in the nervous system to their loss in all somatic tissues we also crossed our AID tagged mTORC1 components to a 'Somatic TIR1' strain that expresses ruby-tagged TIR1 under control of the somatic *eft-3* promoter [23]. These strains are referred to throughout the manuscript as RAGA-1::AID-Somatic, RAGA-1::AID-Neuronal, LET-363::AID-Somatic, and LET-363-Neuronal (Fig 1B).

We tested whether exposure to auxin would lead to degradation of the tagged proteins in our newly generated strains. High concentrations of auxin alone can impact *C. elegans* physiology, such as promoting ER stress resistance [25] and increasing lifespan [26]. We used GFP fluorescence in our RAGA-1::AID-Somatic strain to test whether a lower auxin concentration, 0.15 mM, would be sufficient to degrade RAGA-1::AID::EmGFP. RAGA-1::AID-Somatic animals grown in the absence of auxin have RAGA-1::AID::EmGFP expression in all tissues in the form of green fluorescent puncta (Fig 1D, top panel). We treated RAGA-1::AID-Somatic animals with 0.15 mM auxin beginning at day 1 of adulthood and found that RAGA-1::AID::EmGFP signal was diminished after only 2.5 hours of auxin exposure and was almost completely undetectable by 6 hours (Fig 1D and S1A and S1B Fig). Importantly, this lower dose of auxin does not interfere with *C. elegans* lifespan as we found no change in the lifespan of wild type animals nor in the lifespan of animals expressing the individual AID or TIR1 components alone upon exposure to 0.15 mM auxin (S3A Fig). Thus, 0.15 mM auxin efficiently and rapidly degrades AID-tagged RAGA-1 without incurring negative effects on wild type lifespan.

We further validated the effects of auxin-inducible degradation (AID) of RAGA-1 and LET-363 by assessing phosphorylation levels of a downstream mTORC1 target, RSKS-1 (putative ortholog of mammalian S6K1). mTORC1 inhibition is known to decrease levels of RSKS-1/S6K1 phosphorylation and it has been shown that *raga-1(ok386)* loss-of-function mutants have decreased RSKS-1 phosphorylation [27]. As expected, somatic AID of RAGA-1 or LET-363 resulted in decreased levels of phosphorylated RSKS-1, while auxin treatment had no effect on RSKS-1 levels in wild type animals (S2 Fig). In contrast, neuronal AID of RAGA-1 did not have a significant effect on RSKS-1 phosphorylation as measured from whole animal lysates (S2A and S2B Fig).

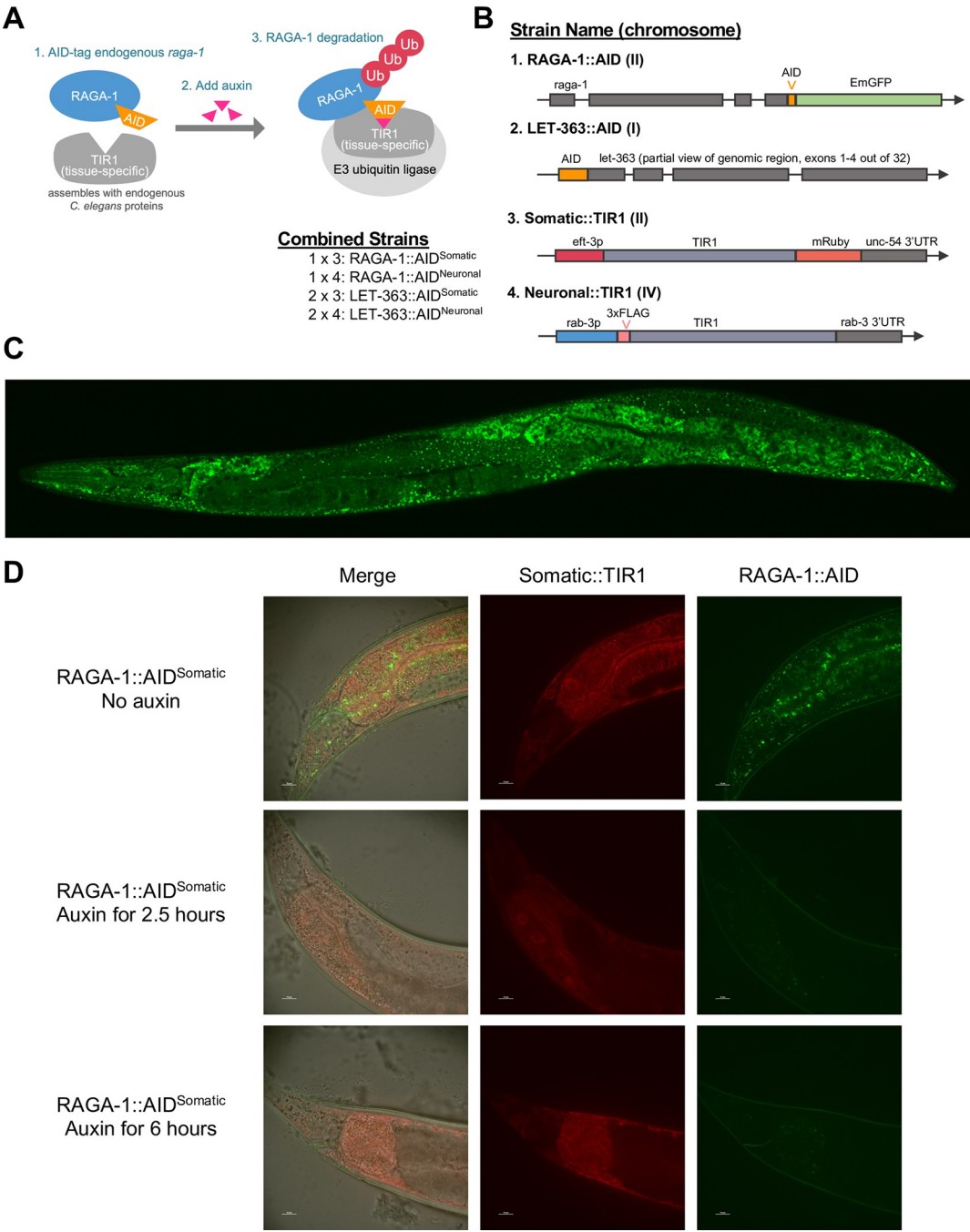

**Fig 1. Auxin inducible degradation (AID) depletes endogenous mTORC1 components in *C. elegans*.** (A) TIR1 assembles with endogenous *C. elegans* proteins to form an E3 ubiquitin ligase that recognizes and degrades AID-tagged proteins in the presence of the chemical auxin. (B) RAGA-1::AID and LET-363::AID were generated by adding the AID degron tag onto endogenous *raga-1* and *let-363* using CRISPR/Cas9 gene editing. The Somatic::TIR1 strain was acquired from Zhang et al. *Development* 2015 and expresses TIR1::mRuby in all somatic tissues. We generated the Neuronal::TIR1 strain to restrict expression of TIR1 to the neurons. These strains were crossed together and are referred to by the names in the "Combined Strains" section of this figure. (C) Expression pattern of endogenous *raga-1* tagged with AID::EmGFP (RAGA-1::AID:: EmGFP). Representative image of 20 animals imaged created by merging 3 images taken at 40x to show expression throughout the whole body. (D) RAGA-1::AID^Somatic worms on control plates have TIR1 expression (red fluorescence) in all somatic tissues (not the germline) and RAGA-1::AID::EmGFP expression (green fluorescence) in all tissues. 2.5 hours after being moved onto plates containing 0.15 mM auxin, TIR1 is still visible but nearly all RAGA-1::AID::EmGFP is degraded in all tissues (14 out of 21 had no visible puncta). After 6 hours on 0.15 mM auxin, the RAGA-1::AID::EmGFP signal is thoroughly diminished. All images shown are of the tail region. n = 2, at least 20 worms imaged per condition.

## Lifespan extension by AID of RAGA-1 in soma and neurons

To test effects of auxin-inducible degradation of RAGA-1 in all somatic tissues and in the nervous system specifically, we performed lifespan assays on the RAGA-1::AID$^{Somatic}$ and RAGA-1::AID$^{Neuronal}$ *C. elegans* strains in the presence and absence of auxin. RAGA-1::AID$^{Somatic}$ animals have a longer median lifespan relative to wild type animals even in the absence of auxin (Fig 2A). When RAGA-1::AID$^{Somatic}$ animals are placed on auxin, median lifespan is not extended but is indeed sometimes shortened. However, maximum lifespan is consistently significantly extended (S3B Fig). The early population crash followed by lifespan extension for the latter portion of the population is reminiscent of the crash often observed in the loss-of-function *raga-1* mutant (S3C Fig) but is more severe in magnitude.

The fact that RAGA-1::AID$^{Somatic}$ animals have a mild lifespan extension even in the absence of auxin might be explained by a low baseline level of RAGA-1 degradation, as mild auxin-independent degradation of AID-tagged proteins has been previously reported [28]. Indeed, even without the addition of auxin, the green fluorescence signal in the RAGA-1::AID$^{Somatic}$ strain is reduced as compared to the signal in the RAGA-1::AID::EmGFP strain (S4 Fig). Thus, the addition of somatically expressed TIR1 is sufficient to induce a low level of auxin-independent degradation of AID-tagged RAGA-1 protein. Upon the addition of auxin, the GFP signal is further diminished down to the level of GFP autofluorescence that is present in wild type animals. In contrast, RAGA-1::AID$^{Neuronal}$ animals don't have a significant decrease in whole-body GFP signal as compared to the RAGA-1::AID::EmGFP strain, with or without auxin, likely due to the persistence of RAGA-1::AID::EmGFP in non-neuronal tissues in the head such as the hypodermis and pharynx (S4 Fig).

Unlike the RAGA-1::AID$^{Somatic}$ strain, RAGA-1::AID$^{Neuronal}$ animals had no difference in median lifespan relative to wild type animals in the absence of auxin. Remarkably, lifelong exposure to auxin significantly extended the lifespan of RAGA-1::AID$^{Neuronal}$ animals (Fig 2B). In *C. elegans*, inhibition of *raga-1* by RNA interference (RNAi) initiated in adult stages is sufficient to extend lifespan [13]. Therefore, we tested whether adult-onset AID of RAGA-1 in the neurons would similarly extend lifespan. RAGA-1::AID$^{Neuronal}$ animals exposed to auxin starting at day 1 of adulthood showed robust extension of lifespan equal to that seen in animals exposed to auxin throughout development (Fig 2B).

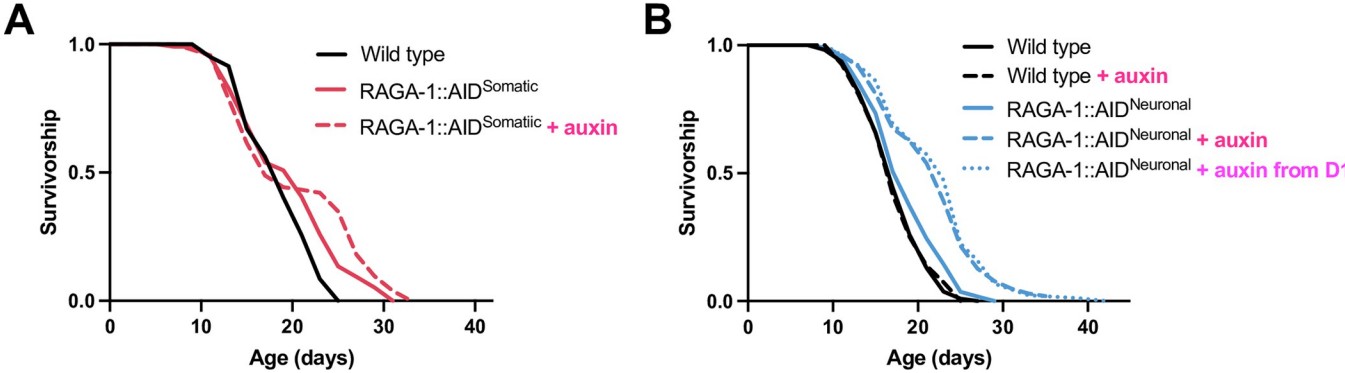

**Fig 2. Neuron-specific inhibition of RAGA-1 extends lifespan in *C. elegans*.** (A) The lifespan curve of RAGA-1::AID$^{Somatic}$ animals with no auxin is significantly different from wild type (p = 0.0232) although they have the same median lifespan (16 days). The lifespan curve of RAGA-1::AID$^{Somatic}$ + auxin (median = 17 days) is significantly different from wild type (p = 0.0005) but not from the RAGA-1::AID$^{Somatic}$ genetic control (p = 0.8528). (B) The lifespan of untreated RAGA-1::AID$^{Neuronal}$ animals is not significantly different from wild type (p = 0.9091). Administering auxin either from hatch or from day 1 of adulthood significantly extends the lifespan of RAGA-1::AID$^{Neuronal}$ animals (p < 0.0001 for both). Complete statistics and lifespan replicates available in **S1 Table**.

One notable example of neuronal control of longevity in *C. elegans* is lifespan extension mediated by the disruption of sensory neuron function, which has been observed through various methods [29,30]. To test whether AID of RAGA-1 in the neurons is extending lifespan through the death or general dysfunction of sensory neurons, we subjected RAGA-1::AID[Neuronal] animals to a panel of chemotaxis assays (S5 Fig). In all cases, the behavior of auxin-treated RAGA-1::AID[Neuronal] animals was indistinguishable from wild type animals, suggesting broad measures of sensory signaling are intact (S5 Fig).

### Degrading RAGA-1 in the neurons uncouples aging from anabolic trade-offs

In addition to lifespan extension, whole body mTORC1 inhibition has negative effects such as impaired growth, slowed development, and decreased reproductive capacity [7,12,13]. Indeed, *raga-1* mutants are slow-growing and small [13]. It has long been assumed that these phenotypes represent energetic trade-offs that are causal to and therefore inseparable from mTORC1 longevity, but this hypothesis has remained untested. Our model of neuron-specific RAGA-1 longevity provides us with an opportunity to test whether mTORC1 regulates any or all of these phenotypes through the neurons and whether these phenotypes are required for neuronal RAGA-1-mediated lifespan extension.

While animals with somatic degradation of RAGA-1 from hatch take significantly longer to develop, animals with neuronal RAGA-1 degradation develop at the same rate as wild type animals (Fig 3A). Similarly, somatic RAGA-1 degradation significantly reduces egg production, yet RAGA-1 degradation causes no impairment in fecundity (Fig 3B). Finally, animals with somatic AID of RAGA-1 from hatch are significantly shorter on day 1 of adulthood

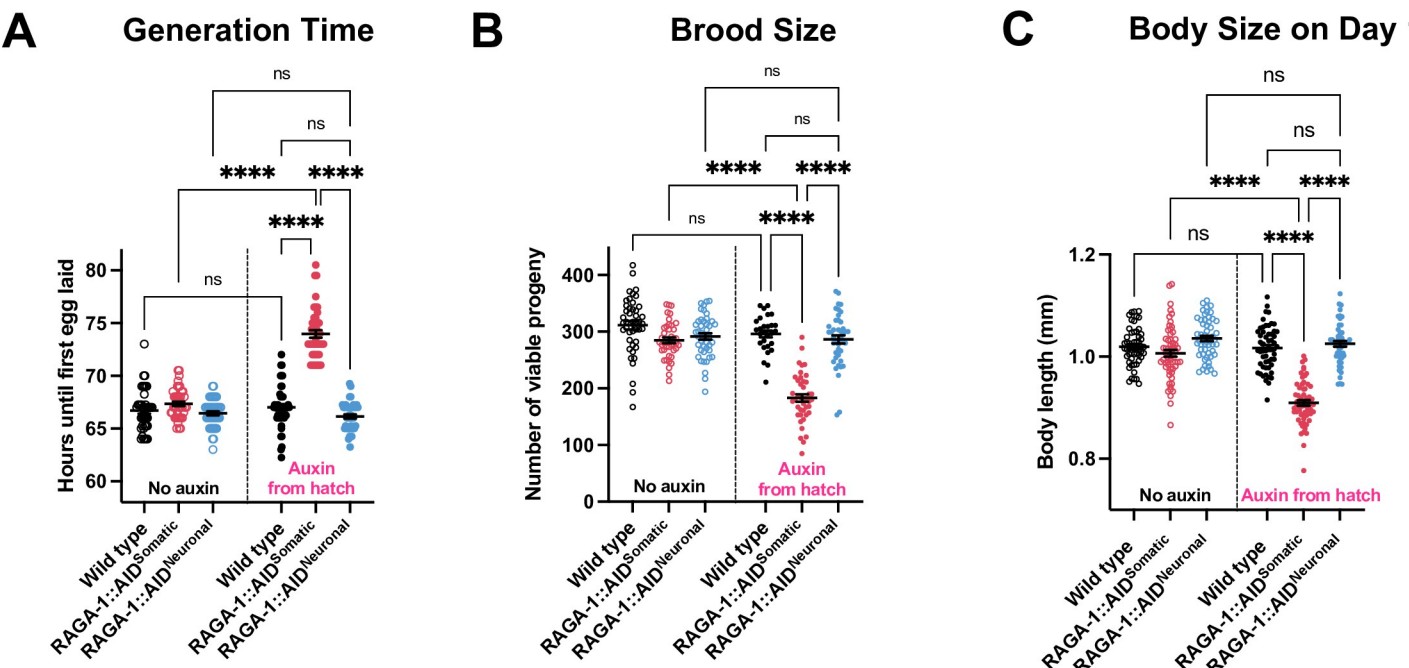

**Fig 3. Growth, reproduction, and development of animals following somatic or neuronal AID of RAGA-1.** (A) Animals with somatic loss of RAGA-1 take, on average, 7 hours longer to develop into egg-laying adults while animals with neuronal loss of RAGA-1 do not have a developmental delay. n = 3 with at least 44 individual measurements per condition. (B) Animals with somatic degradation of RAGA-1 results have fewer progeny while animals with neuronal degradation of RAGA-1 have no impairment in reproduction. n = 3 with at least 43 individual worms for all conditions except for "Wild type + auxin" which has n = 2 with a total of 29 individual measurements. (C) Body length of animals measured on day 1 of adulthood. There is no difference in body size between any of the strains when untreated with auxin. RAGA-1::AID[Somatic] animals are significantly smaller when grown on auxin while the RAGA-1::AID[Neuronal] strain has no growth defect following auxin treatment. n = 3 independent replicates with a total of at least 52 animals measured per condition. Each point indicates a measurement from one animal. Error bars are plotted at mean ± SEM. **** indicates p <0.0001.

than wild type animals, but neuronal AID of RAGA-1 has no impact on body size (Fig 3C). We also measured body size and development time of auxin-treated strains expressing AID:: EmGFP-tagged RAGA-1, Neuronal::TIR1, and Somatic::TIR1 alone to test whether individual components of the AID system would impact the phenotypes we assayed. We find that auxin-treated RAGA-1::AID and Neuronal::TIR1 animals have wild type body size and generation time while animals with Somatic::TIR1 expression have a slight increase in body size and generation time (S6 Fig). Although this suggests that Somatic::TIR1 has mild effects on growth and development, the key phenotype of interest in the auxin-treated RAGA-1::AID^Somatic animals is a decrease in body size and generation time. Therefore, there is no evidence to suggest that our results are confounded by the effects of Somatic::TIR1.

We further assessed the effects of somatic and neuronal RAGA-1 degradation on growth during the developmental stages using an automated microchamber imaging setup as well as TIR1 expression driven by a different pan-neuronal promoter, *rgef-1*. Animals with somatic AID of RAGA-1 take significantly longer to reach each larval molt and have a slower growth rate throughout development (S7 Fig). In contrast, animals with neuronal AID of RAGA-1 have no defect in larval stage duration or overall growth rate (S7 Fig). Thus, we have driven TIR1 expression using two distinct pan-neuronal promoters and measured growth during both development and adulthood to find that neuronal RAGA-1 depletion has no effect on whole animal growth.

These findings have multiple implications for mTORC1 longevity in *C. elegans*. First, that animals with neuronal AID of RAGA-1 are long-lived but have normal growth, development, and reproduction demonstrates that these phenotypes are indeed separable from longevity. Second, these data shed new light on tissue-specific functions of RAGA-1; while RAGA-1 can modulate aging from the neurons of *C. elegans*, it must regulate whole body size, development rate, and reproductive capacity through non-neuronal tissues. Altogether, this work demonstrates that neuron-specific mTORC1 inhibition not only extends lifespan but also does so without disrupting anabolic processes such as organismal growth, development, and reproduction.

## Neuronal AID uncouples aging and growth phenotypes resulting from loss of LET-363/mTOR

Our finding that RAGA-1 regulation of lifespan can be uncoupled from other anabolic phenotypes suggest that this could be true for other components of mTORC1 signaling as well. Therefore, we analyzed the lifespan and growth of animals following neuronal and somatic AID of LET-363/mTOR. Previously, it has been difficult to assess the lifespan effects of LET-363 manipulation because loss of *let-363* results in larval arrest [31]. Additionally, partial knockdown of *let-363* with RNA interference (RNAi) leads to confounding results as the RNAi also knocks down a mitochondrial gene, *mrpl-47*, that exists in an operon with *let-363* and has been found to affect lifespan [32,33]. Therefore, the tissue-specificity and inducibility of the AID system is particularly advantageous here and allows us to assess the relationship between LET-363 and aging with unprecedented specificity.

AID of LET-363 in all somatic tissues from hatch results in larval arrest as previously reported with *let-363* knockdown [31] (Fig 4A). To avoid larval arrest, we tested the effect on lifespan of somatic or neuronal LET-363 degradation beginning in adulthood. Whole body degradation of LET-363 from day 1 of adulthood onwards causes a large proportion of the population to die by bagging, an egg-laying defect in which the larvae hatch inside of the adult (S1 Table). The small body size resulting from LET-363 degradation likely contributes to this phenomenon (Fig 4C and S8 Fig). The LET-363::AID^Somatic animals that are not lost to bagging have a severely shortened median lifespan (Fig 4B). However, degradation of LET-363 only in the neurons extends the lifespan of *C. elegans* (Fig 4B) without causing the bagging

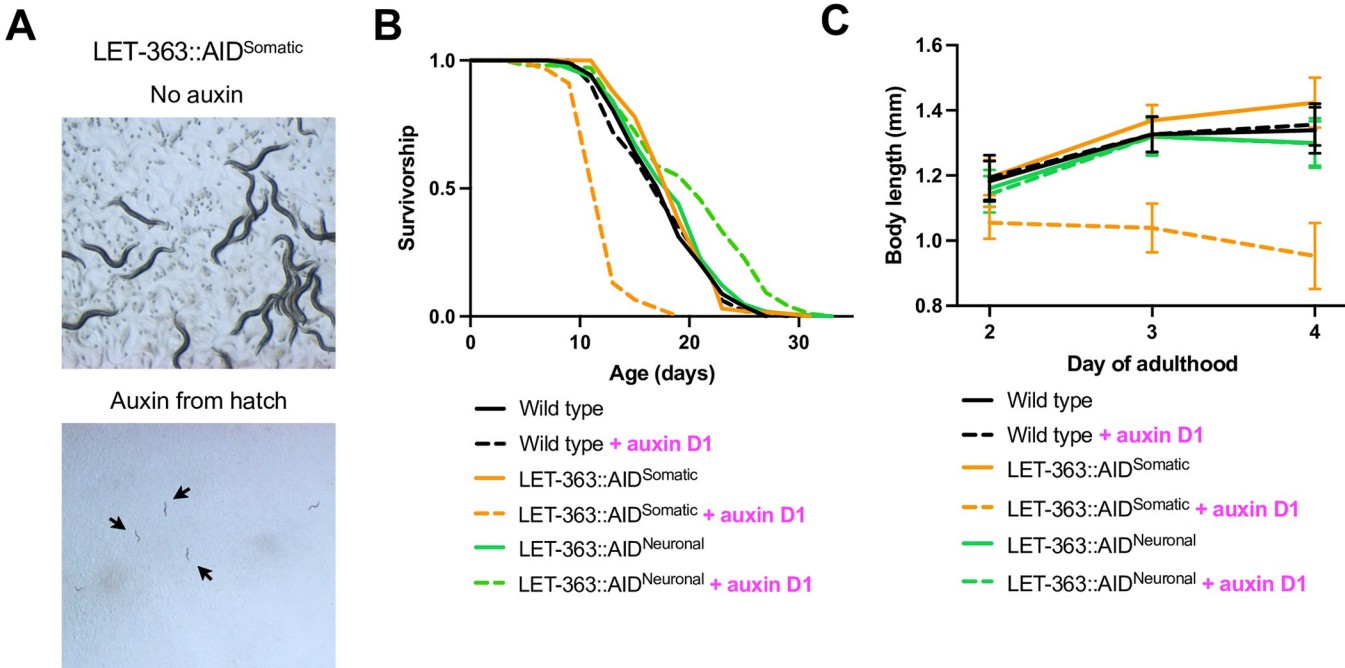

**Fig 4. Neuronal AID of LET-363/mTOR extends lifespan without impairing growth.** (A) Auxin from hatch causes LET-363::AID^Somatic worms to arrest at the larval L3 stage. (B) Adult-onset auxin exposure has no effect on wild type worms (p = 0.8115) but shortens the lifespan of LET-363::AID^Somatic worms (p < 0.0001). It is important to note that the majority of the LET-363::AID^Somatic animals were censored due to bagging, see **S1 Table**. On the contrary, adult-onset auxin exposure extends the lifespan of LET-363::AID^Neuronal worms (p = 0.0011). n = 3. (C) Animals were treated with auxin beginning on day 1 of adulthood and body length was measured on days 2, 3, and 4. LET-363::AID^Somatic animals treated with auxin are significantly smaller on all days but auxin treatment doesn't affect body size in wild type animals. Auxin-treated LET-363::AID^Neuronal animals have a mild but significant (4%) decrease in body size on days 2 and 4. n = 3 independent replicates with a total of at least 58 individual worms measured for each condition. Points represent the mean and standard deviation of all animals measured in 1 biological replicate at that time point. Full statistics in **S8 Fig**.

defect observed upon ubiquitous LET-363 degradation. These data demonstrate that while whole-body LET-363 degradation is deleterious, targeting LET-363 only in the neurons is advantageous for promoting longevity. Thus, LET-363/mTOR can have both pro-longevity and deleterious phenotypes dependent upon the tissue in which its activity is modulated. This highlights the importance of identifying the key tissue(s) in which a pathway acts to regulate aging rather than targeting it broadly throughout the entire body.

In addition to extending rather than shortening lifespan, restricting LET-363 degradation to the neurons remedies the severe growth impairment observed with somatic LET-363 loss. We measured body size following adult-onset somatic and neuronal LET-363 degradation. At 24-, 48-, and 72-hours post onset of auxin treatment, somatic LET-363 degradation significantly decreases body size resulting in a 29.7% growth reduction by 72 hours compared to wild type animals. In contrast, neuronal LET-363 degradation did not cause a drastic size reduction, with only a 4% decrease in size compared to auxin-treated wild type animals observed at 24 and 72 hours (Fig 4C and S8 Fig).

## Differential gene expression following somatic RAGA-1 AID reflects mTORC1 inhibition

Our model of neuron-specific mTORC1 inhibition reveals that many of the phenotypes induced by whole-body mTORC1 inhibition are not required for mTORC1 mediated lifespan extension. We hypothesized that by comparing changes in gene expression following either

neuronal or somatic RAGA-1 degradation, we could identify tissue-specific mTORC1 functions that are causal for longevity.

We performed RNA-sequencing of RNA extracts from whole worm lysates following adult-onset degradation of RAGA-1 in either all somatic tissues or only in the neurons. Principal component analysis reveals that all "no auxin" conditions cluster together, demonstrating that the strains are relatively similar at baseline despite differences in their genetic background (Fig 5A). The "wild type + auxin" condition also clustered with the untreated conditions, demonstrating that auxin alone does not have a significant effect on gene expression. The neuronal AID and somatic AID conditions cluster distinctly, with the somatic AID condition clustering farthest away from the rest of the samples (Fig 5A). This fits with our expectation that degradation of RAGA-1 in all tissues would cause the largest perturbation in global gene expression.

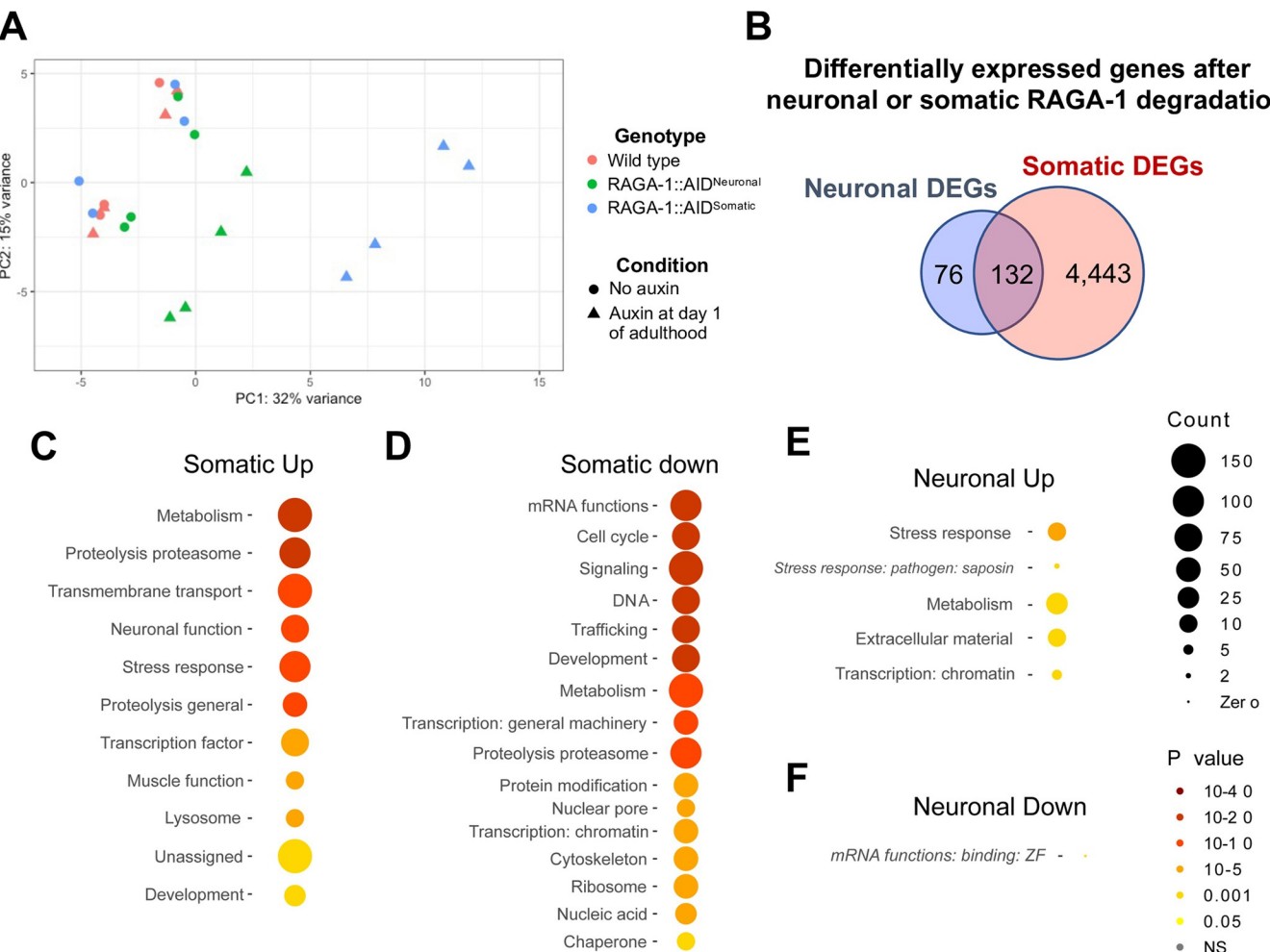

**Fig 5. RNA-Sequencing of animals following somatic or neuronal AID of RAGA-1.** Animals were placed on auxin or control plates at day 1 of adulthood and RNA was harvested on day 3 of adulthood and sent for sequencing. n = 4 independent replicates with each condition containing 400 animals per replicate. (A) PCA plot of all conditions sequenced show that RAGA-1::AID^Somatic and RAGA-1::AID^Neuronal animals treated with auxin cluster distinctly and away from the experimental control strains. (B) Venn diagram of significantly differentially expressed genes (DEGs) between auxin-treated and untreated conditions. (C-F) Analysis with the *C. elegans*-specific gene set enrichment analysis tool, WormCat. Functional categories enriched in DEGs upregulated (C) or downregulated (D) after somatic AID of RAGA-1. Functional categories enriched in DEGs upregulated (E) or downregulated (F) after neuronal AID of RAGA-1.

Somatic AID of RAGA-1, even for only 48 hours, resulted in significant differential expression of 4,575 genes (S2 Table). In contrast, neuronal RAGA-1 degradation only significantly altered the expression of 208 genes (Fig 5B and S2 Table), suggesting a more specific effect of neuronal mTORC1 inhibition as opposed to ubiquitous inhibition. The overlapping 132 genes between the two differentially expressed gene (DEG) sets may represent a core set of genes that regulate aging following either somatic or neuronal mTORC1 inhibition.

Using WormCat, a gene set enrichment analysis tool specific to *C. elegans*, we find that the differentially expressed genes following somatic AID of RAGA-1 contain canonical signatures of mTORC1 inhibition: lipid metabolism, amino acid breakdown, E3 proteasomal proteolysis and neuronal signaling are upregulated while mRNA processing, ribosome biogenesis, somatic development, cell cycle, and small GTPase signaling are downregulated (Fig 5C and 5D). We also find changes in specific genes that have previously been shown to be regulated by mTORC1 inhibition. For example, many of the top upregulated genes belong to the SKN-1 dependent zygotic (sdz) gene class and it has previously been shown that mTORC1 inhibition results in increased expression of SKN-1 target genes [7]. Additionally, *pha-4*, a downstream effector of mTORC1 longevity, is significantly upregulated 3.34-fold and *pha-4* expression has been shown to increase following RNAi against *let-363* in *C. elegans* [34].

## Neuronal RAGA-1 degradation drives differential expression of stress response and metabolism genes

DEGs upregulated following neuronal RAGA-1 degradation are enriched for genes involved in stress response, metabolism, extracellular material, and transcription (chromatin) (Fig 5E). The only functional category in the group of 33 downregulated genes is mRNA-binding zinc fingers, and this group only contained 2 genes (Fig 5F). Although many of the DEGs are expressed in both neurons as well as other tissues, we find that neuronal RAGA-1 degradation induces differential expression of non-neuronal genes—such as *plin-1*, *gnsl-1*, *myo-2*, and *ges-1* –suggesting that neuronal mTORC1 is able to cell nonautonomously regulate gene expression in peripheral tissues.

Notably, unlike in the DEGs from animals with somatic RAGA-1 degradation, we do not observe an enrichment in neuron-specific genes or differential expression of neuropeptides following neuronal RAGA-1 degradation. Only 4 genes categorized to have neuron-specific functions are present in our neuronal DEG dataset, and all 4 are upregulated: *cat-4* (biosynthesis of dopamine, melanin, and tetrahydrobiopterin), *frpr-5* (a predicted G protein coupled receptor with no known ligand), *cab-1* (involved in synaptic transmission), and *tom-1* (synaptic vesicle trafficking). We also find that 6 genes with unassigned functions that are known to be enriched in dopaminergic neurons are upregulated, suggesting a possible role of this specific subset of neurons in mediating neuronal RAGA-1 longevity.

Finally, we investigated whether our RNA sequencing data supported our finding that restricting RAGA-1 degradation to the neurons uncouples longevity from the anabolic trade-offs typically associated with whole-body mTORC1 inhibition (Figs 2 and 3). Indeed, somatic RAGA-1 degradation results in downregulation of many genes involved in cell cycle, development, and DNA replication. In contrast, these genes are unaffected by neuron-specific RAGA-1 degradation, potentially explaining why animals with neuronal loss of RAGA-1 do not have the same growth, developmental, and reproduction impairments as animals with somatic RAGA-1 loss. Neuron-specific RAGA-1 degradation results in differential expression of very few genes relative to somatic RAGA-1 degradation yet is sufficient to extend lifespan. The limited changes in gene expression support our finding that targeting mTORC1 neuronally uncouples longevity from anabolic trade-offs and underscores stress response and metabolism as key pathways involved in mTORC1 regulation of aging in *C. elegans*.

### Testing requirement of factors previously implicated in mTORC1 longevity

Given that *pha-4* has previously been shown to be required for longevity resulting from *let-363* inhibition, we tested whether it might also be required for longevity mediated by neuronal RAGA-1 degradation. Neuronal degradation of RAGA-1 is still able to extend the lifespan of animals on *pha-4* RNAi, demonstrating that *pha-4* is not required for this longevity paradigm (Fig 6A). Overall, the vast changes in gene expression observed following AID of RAGA-1 in the soma align with previous studies on mTORC1 inhibition, but many of these genes, such as *pha-4*, are likely not required for longevity resulting from tissue-specific RAGA-1 manipulation.

In addition to *pha-4*, splicing factor 1 (*sfa-1*) was also identified as significantly upregulated (by 1.27-fold) following somatic degradation of RAGA-1. SFA-1 was previously found to be required for longevity of the *raga-1*(ok386) loss-of-function mutant in *C. elegans* [27]. Therefore, we assessed the requirement of SFA-1 for the longevity resulting from neuronal RAGA-1 degradation. We find that RNAi against *sfa-1* suppresses the lifespan extension of animals with neuronal RAGA-1 degradation (Fig 6B), suggesting that the longevity resulting from neuron-specific RAGA-1 loss is mediated in part by the same mechanisms as whole-body RAGA-1 manipulation. It will be interesting to uncover whether required factors, such as SFA-1, are required cell autonomously in the neurons or, rather, are contributing to longevity through peripheral tissues. This model of tissue-specific RAGA-1 longevity provides us with a unique opportunity to uncover how a pathway targeted in one tissue then regulates downstream pathways cell autonomously and cell nonautonomously to ultimately promote lifespan extension.

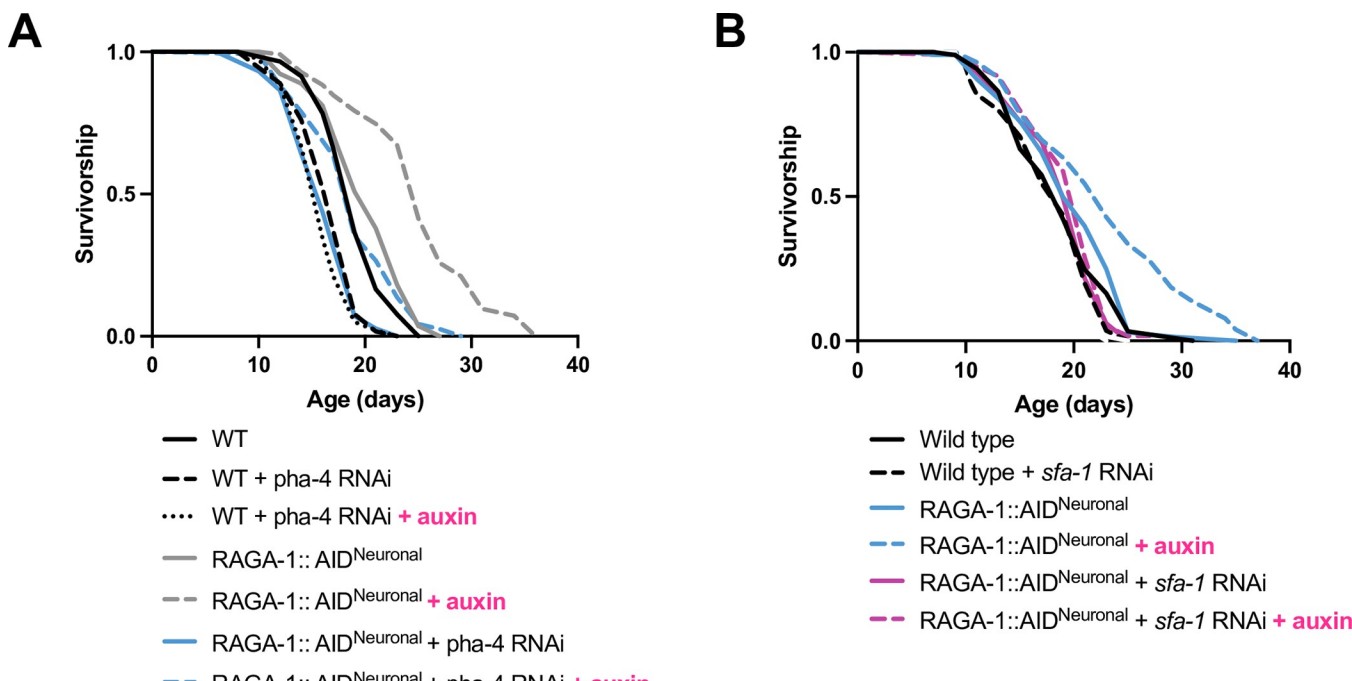

**Fig 6. Requirement of PHA-4 and SFA-1 for neuronal RAGA-1 longevity.** (A) Neuronal RAGA-1 degradation is able to extend lifespan of animals on *pha-4* RNAi (p < 0.0001 for "RAGA-1::AID$^{Neuronal}$ + pha-4 RNAi" vs "RAGA-1::AID$^{Neuronal}$ + pha-4 RNAi + Auxin). n = 3. (B) Neuronal RAGA-1 degradation is unable to extend lifespan when the splicing factor *sfa-1* is depleted with RNA interference ("RAGA-1::AID$^{Neuronal}$ + sfa-1 RNAi" vs. "RAGA-1::AID$^{Neuronal}$ + sfa-1 RNAi + auxin", p = 0.2026), indicating that SFA-1 is required for longevity. Animals in conditions "+ auxin" were grown on plates containing 0.15 mM auxin for their whole lives (from hatch). n = 2. Complete statistics and lifespan replicates available in **S1 Table**.

## Discussion

Decades of research into the identification of biological pathways that regulate the rate of aging have converged on conserved nutrient metabolism pathways such as mTORC1. Several studies have revealed that inhibition of the mTORC1 pathway is a promising strategy to promote healthy aging. However, translation of mTORC1 therapeutics into humans is hindered by the fact that whole-body mTORC1 inhibition also disrupts its role in promoting anabolic metabolism and induces undesirable effects such as reduced fertility, stunted growth, metabolic disruption, and immunosuppression. One potential strategy to circumvent these negative effects is to identify the key tissue(s) in which mTORC1 acts to regulate aging and then develop targeted therapies that promote healthy aging while minimizing undesirable off-target effects of global mTORC1 inhibition.

Here, we find that neuron-specific mTORC1 inhibition is sufficient to promote longevity in *C. elegans* using AID, a robust system with multiple advantages over methods that have previously been used to study tissue-specific effects of mTORC1 on aging. In *C. elegans*, tissue-specific RNAi of *ragc-1* in the intestine [7] or *ifg-1* in the germline, hypodermis, or neurons [22] has been shown to extend lifespan. In *Drosophila melanogaster*, lifespan was extended by expression of mTORC1 pathway components in the fat tissue using the GAL4-UAS system [8]. However, further research has shown that both tissue-specific RNAi and GAL4-UAS drivers [35] can have off-target effects in other tissues. In addition to being more tissue-specific than RNAi, AID has the added benefit of targeting the protein directly rather than silencing a gene at the mRNA level. Therefore, late-onset AID is able to eliminate any of the target protein that is present while late-onset RNAi prevents translation of new protein but is unable to eliminate existing protein. Researchers have taken advantage of this aspect of the AID system to show that late-life degradation of DAF-2, the *C. elegans* ortholog of the mammalian insulin and IGF receptor, extends lifespan [36]. Additionally, two studies have found that AID of DAF-2 in only the intestine or the neurons is sufficient to extend lifespan [36,37]. The inducibility and tissue-specificity of the AID system has expanded our understanding of the temporal and tissue requirements for nutrient metabolism pathways in the regulation of aging.

To begin to understand the mechanisms of neuronal RAGA-1 longevity, we tested the requirement for two factors associated with mTORC1 longevity: the splicing factor SFA-1 which is required for longevity in the *raga-1(ok386)* mutant and the transcription factor PHA-4 which is required for longevity resulting from reduced LET-363/mTOR. We found that SFA-1, but not PHA-4, is required for neuronal raga-1 lifespan extension, demonstrating that this longevity intervention may only require a subset of factors thought to be necessary for mTORC1 longevity.

Our finding that somatic AID of RAGA-1 doesn't extend median lifespan was somewhat surprising but, in fact, this is the first study of the lifespan effect of complete depletion of RAGA- in *C. elegans*. Previous studies showing that *raga-1* loss extends lifespan rely on either partial knockdown using RNAi or loss-of-function mutants in which some of the coding region is still intact [7,12,13]. Indeed, Schreiber et al. analyzed the lifespan of 3 different *raga-1* loss-of-function mutants. *raga-1(ok386)* has a deletion beginning near the start of the 2nd exon and was long-lived [12], as has been confirmed in other studies [13]. *raga-1(ok701)* has a smaller deletion, beginning halfway through exon 2, and has an even longer lifespan extension than *raga-1(ok386)*. Finally, *raga-1(tm1862)* has the largest deletion, beginning in the middle of exon 1, and was not significantly long-lived [12]. Mammalian data also supports the notion that complete loss of mTORC1 components throughout the body is deleterious as RagA loss in mice is lethal [38]. These data, along with our data on somatic *raga-1* degradation, suggest that the degree of *raga-1* depletion is a crucial determinant of lifespan outcomes. It is possible that

whole-body mTORC1 inhibition could be fine-tuned to maximize lifespan extension while minimizing deleterious side-effects by carefully controlling the level of inhibition, for example by performing AID on *C. elegans* heterozygous for RAGA-1::AID or LET-363::AID. Moreover, while we believe that the longevity resulting from neuronal AID of RAGA-1 is due in part to tissue-specific functions of the mTORC1 pathway, the overall level of mTORC1 pathway activity might also play a role.

The AID system allowed us to interrogate the effects of adult-onset somatic and neuronal degradation of LET-363/mTOR, a protein that has been difficult to study with interventions at the genome and transcript level because its deletion is lethal. Unlike somatic AID of RAGA-1, adult-onset AID of LET-363 in all somatic tissues results in a rapid die-off of the population. The more severe phenotypes associated with somatic loss of LET-363 could be explained by the fact that LET-363/mTOR is also a subunit of mTOR complex 2, so manipulation of LET-363 affects both mTORC1 and mTORC2 signaling. In contrast, neuron-specific AID of LET-363, like neuronal AID of RAGA-1, extends median lifespan.

Our finding that neuronal mTORC1 inhibition extends lifespan without impairing growth, development, or fertility challenges many of the current theories about the mechanisms driving mTORC1 longevity. It has long been assumed that anabolic trade-offs are causal to mTORC1 longevity because the result is the redirection of resources from growth and reproduction back into somatic maintenance. Indeed, there is a strong association between small body size and longevity that is not unique to mTORC1 inhibition. Within a species, smaller individuals tend to live longer [39–43] and both animals and humans with mutations in the IIS pathways are smaller and tend to have longer lifespans [44,45]. However, we find that animals with neuron-specific mTORC1 inhibition are long-lived yet are the same size and have the same reproductive capacity as wild type animals, thereby uncoupling longevity from anabolic trade-offs. Furthermore, genes involved in cell cycle and development are differentially expressed following somatic RAGA-1 degradation but unchanged by neuronal RAGA-1 degradation. Altogether, these data suggest that impaired reproduction, development, and whole-body growth are driven by mTORC1 inhibition in non-neuronal tissues, and these phenotypes are not required for mTORC1 longevity.

This work is the first demonstration of neuron-specific manipulation of mTORC1 or its upstream regulator being sufficient to promote lifespan extension. In support of the idea that aging can be regulated through manipulation of nutrient metabolism pathways in the neurons, inhibition of the IIS pathway in neurons has been shown to extend lifespan in *C. elegans* [46], *Drosophila melanogaster* [47], and mice [48,49] and knockdown of a downstream mTORC1 target in the neurons extends lifespan in *C. elegans* [22]. Previous work from our lab has shown that the energy sensory AMPK is required in the neurons for lifespan extension by mTORC1 inhibition [13]. This finding along with the results presented here raises the interesting possibility that neuronal AMPK cooperates cell autonomously with neuronal mTORC1 to regulate aging. Additionally, in *C. elegans*, direct manipulation of the sensory neurons has been shown to extend lifespan [29,30]. We find that neuronal mTORC1 inhibition does not appear to broadly impair sensory neuron function and is likely mediating longevity through a distinct mechanism. It is possible that mTORC1 inhibition is not required pan-neuronally to achieve longevity. Indeed, we find that neuronal RAGA-1 degradation results in upregulation of multiple genes enriched in dopaminergic neurons as well as a gene involved in dopamine biosynthesis, pointing to a possible link between mTORC1 signaling and dopaminergic neurons.

Previous work from our lab found that neuronal vesicle release is required for mTORC1 to regulate aging from the neurons, suggesting a role for neuronal signaling and cell nonautonomous communication in neuronal mTORC1 longevity. Indeed, we find changes in the expression of genes involved in neuronal signaling and the expression of non-neuronal genes

following neuronal RAGA-1 degradation, suggesting that manipulation of neuronal mTORC1 alters neuronal communication and has cell nonautonomous effects in peripheral tissues. Additionally, somatic RAGA-1 degradation altered the expression ~50 neuropeptide genes, none of which were altered when RAGA-1 was degraded in the neurons alone, suggesting that mTORC1 also participates in periphery-to-neuron signaling.

This research demonstrates the importance of determining whether conserved longevity pathways act throughout the organism or only in specific tissues to regulate aging. By identifying that mTORC1 inhibition in the neurons is sufficient to extend lifespan, we are able to use a more specific intervention to limit off-target effects of mTORC1 pathway manipulation and identify the downstream changes, like stress response and metabolism, that are more likely causal to longevity. Future studies on neuronal mTORC1 inhibition and neuron-specific functions of mTORC1 are imperative for the advancement of mTORC1-centric anti-aging therapeutic strategies.

## Materials and methods

### *C. elegans* strains and husbandry

Worms were maintained at 20˚C on nematode growth media (NGM) plates with Carbenicillin seeded with *E. coli* strain HT115 (CGC) using standard techniques [50]. *E. coli* were grown in LB in a shaker at 37˚C overnight, then 100 μL of the liquid culture was seeded onto the plates and left for 2 days to grow at room temperature.

### *C. elegans* strains Table

| Strain | Genotype | Notes* | Source |
|---|---|---|---|
| N2 Bristol | Wild type | | CGC |
| WBM1204 | *raga-1(wbm40) [raga-1::degron::EmGFP] II* | **RAGA-1::AID** | Mair Lab |
| WBM1481 | *let-363(wbm46) [degron::let-363] I* | **LET-363::AID** | Mair Lab |
| WBM1223 | *wbmIs83 [rab-3p::3xflag::TIR1::rab-3 3'UTR *wbmIs66] IV* | **Neuronal::TIR1** | Mair Lab |
| CA1200 | *ieSi57 [eft-3p::TIR1::mRuby::unc-54 3'UTR + Cbr-unc-119(+)] II; unc-119(ed3) III* | **Somatic::TIR1** | CGC |
| WBM1395 | *ieSi57 [eft-3p::TIR1::mRuby::unc-54 3'UTR + Cbr-unc-119(+)] II* | CA1200 backcrossed 6 times to N2 | Mair Lab |
| WBM1438 | *raga-1(wbm40) [raga-1::AID::EmGFP], ieSi57 [eft-3p::TIR1::mRuby::unc-54 3'UTR + Cbr-unc-119(+)] II* | **RAGA-1::AID**[Somatic], WBM1204 x WBM1395, | Mair Lab |
| WBM1250 | *raga-1(wbm40) [raga-1::degron::EmGFP] II; wbmIs83 [rab-3p::3xflag::TIR1::rab-3 3'UTR *wbmIs66] IV* | **RAGA-1::AID**[Neuronal], WBM1204 x WBM1223 | Mair Lab |
| WBM1490 | *let-363(wbm46) [AID::let-363] I; ieSi57 [eft-3p::TIR1::mRuby::unc-54 3'UTR + Cbr-unc-119(+)] II* | **LET-363::AID**[Somatic], WBM1481 x WBM1395 | Mair Lab |
| WBM1491 | *let-363(wbm46) [AID::let-363] I; wbmIs83 [rab-3p::3xflag::TIR1::rab-3 3'UTR *wbmIs66] IV* | **LET-363::AID**[Neuronal], WBM1481 x WBM1223 | Mair Lab |
| PR811 | *osm-6(p811) V* | **osm-6 sensory mutant** | CGC |
| wBT60 | *raga-1(wbm40 [raga-1:: aid::gfp]) II; bqSi577 [myo-2p::gfp] IV; wbmIs88 [eft-3p::3xflag::dpy-10 crRNA::sl2::wrmscarlet::unc-54 3' UTR] V* | **No TIR1 in Fig 3 –S2** | Towbin Lab at Univ. of Bern, Switzerland |
| wBT188 | *reSi7 [rgef-1p::tir1::f2a::mtagbfp2::nls::aid::tbb-2 3'UTR] I; raga-1(wbm40 [raga-1:: aid::gfp]) II; bqSi577 [myo-2p::gfp] IV; wbmIs88 [eft-3p::3xflag::dpy-10 crRNA::sl2::wrmscarlet::unc-54 3' UTR] V* | **rgef-1p::TIR1 in Fig 3 –S2** | Towbin Lab at Univ. of Bern, Switzerland |
| wBT190 | *raga-1(wbm40 [raga-1::aid::gfp]) II.; xeSi376[eft-3p::tir1::mruby::unc-54 3'UTR] III.; bqSi577 [myo-2p::gfp] IV.; wbmIs88 [eft-3p::3xflag::dpy-10 crRNA::sl2::wrmscarlet::unc-54 3' UTR] V* | **eft-3p::TIR1 in Fig 3 - S2** | Towbin Lab at Univ. of Bern, Switzerland |

*All Mair Lab strains are 6x outcrossed to N2 Bristol unless otherwise stated

## Microinjection and CRISPR/Cas9 gene editing

CRISPR-mediated gene editing was performed according to [51]. For the Neuronal::TIR1 strain, the TIR1 CRISPR mix was injected into the Neuronal SKILODGE strain as described in [24]. Briefly, homology repair templates were amplified by PCR, using primers that introduced a minimum stretch of 35 bp homology at both ends. Single-stranded oligo donors (ssODN) were also used as repair templates. CRISPR injection mix reagents were added in the following order: 0.375 μl Hepes pH 7.4 (200 mM), 0.25 μl KCl (1 M), 2.5 μl tracrRNA (4 μg/μl), 0.6 μl *dpy-10* crRNA (2.6 μg/μl), 0.25 μl *dpy-10* ssODN (500 ng/μl), and PCR or ssODN repair template(s) up to 500 ng/μl final in the mix. If the edit was on Chromosome II, *dpy-5* was used as a coinjection marker. Water was added to reach a final volume of 8 μl. 2 μl purified Cas9 (12 μg/ μl) was added at the end, mixed by pipetting, spun for 2 min at 13000 rpm and incubated at 37˚C for 10 min. Mixes were microinjected into the germline of day 1 adult hermaphrodite worms using standard methods [52]. Alleles were verified by PCR and strains and most strains were backcrossed to N2 wild type worms 6 times before use to eliminate off-target mutations. Strain outcross information can be found in the table above.

## Auxin-inducible degron system

TIR1 was amplified from the plasmid pLZ31; pLZ31 (pCFJ151_Peft-3_TIR1_linker_mRuby_unc-54 3'UTR) was a gift from Abby Dernburg (Addgene plasmid # 71720; http://n2t.net/ addgene:71720; RRID:Addgene_71720). The AID degron tag was amplified from the plasmid pLZ29; pLZ29 (pCFJ151_Peft-3_degron_EmGFP_unc-54 3'UTR) was a gift from Abby Dernburg (Addgene plasmid # 71719; http://n2t.net/addgene:71719; RRID:Addgene_71719).

## Pouring auxin and control plates

To make the 400 mM stock of auxin, indole-3-acetic acid (98+%, A10556-06, Thermo Scientific) was prepared in ethanol, filter-sterilized, and stored at 4˚C in a foil-covered tube as described in [23]. A batch of NGM (with or without Carbenicillin) was prepared and then divided. In one half, 400 mM auxin stock was added to bring the final concentration to 0.15 mM auxin. To the other half, an equivalent amount of filter sterilized 100% ethanol was added as a control for the "No auxin" plates. As auxin is light-sensitive, auxin and control plates were covered in a black tarp or foil to dry overnight before being moved to 4˚C. Plates were taken out and covered one night prior to seeding with a culture of HT115 bacteria.

## Imaging of somatic RAGA-1::AID::EmGFP degradation

RAGA-1::AID[Somatic] animals were grown on NGM Carb plates until day 1 of adulthood (3 days after hatch). On day 1, some animals were taken to be imaged before auxin treatment. Of the remaining animals, half of the animals were transferred to control plates and half were transferred to 0.15 mM auxin plates. Animals were then imaged at 2.5 hours or 6 hours to monitor RAGA-1::AID::EmGFP degradation.

For imaging, worms were anesthetized in 0.5 mg/ml tetramisole in 1X M9 buffer on empty NGM plates and mounted on thin 2% agarose pads on glass slides with 0.05 mm Polybead microspheres (Polysciences) for immobilization. A No. 1.5 cover glass was gently placed on top of worms and sealed with clear nail polish. Images were performed on a Yokogawa CSU-X1 spinning disk confocal system (Andor Technology, SouthWindsor, CT) with a Nikon Ti-E inverted microscope (Nikon Instruments,Melville, NY), using a Plan-Apochromat 100x/ 1.45 objective lens. Images were acquired using a Zyla cMOS camera and NIS elements software was used for acquisition parameters, shutters, filter positions and focus control.

## Imaging of RAGA-1::AID::EmGFP with and without TIR1

Animals were grown on either control or 0.15 mM auxin plates from hatch. On day 1 of adulthood, worms were anesthetized with 0.5 mg/mL tetramisole in M9 on an unseeded plate then mounted on a 2% agarose pad with 0.05 mm Polybead microspheres (Polysciences) for immobilization. A No. 1.5 cover glass was gently placed on top of the worms. Images were taken using a Zeiss Imager.M2 microscope at 20x using consistent exposure levels across all conditions. Image analysis was performed using Fiji software by outlining the entire worm and calculating mean GFP intensity for each animal.

## Western blotting of phosphorylated RSKS-1

To collect protein, 200–400 animals per condition were pelleted in M9 buffer and snap frozen in liquid nitrogen. Pellets were reconstituted in RIPA buffer with protease inhibitor cocktail (Sigma #8340) and phosphatase inhibitor (Roche 4906845001) then lysed by sonication (Qsonica Q700). Normalized protein lysates were loaded onto 10% Tris-Glycine gels (Thermo Fisher Scientific, #XP00100). Proteins were transferred to nitrocellulose membranes (Bio-Rad 1620112) and blocked with 5% milk in TBST. Primary antibodies were added in 5% BSA in milk and incubated overnight at 4°C at the following dilutions: phospho-Drosophila p70 S6 Kinase (Thr398) (Cell Signaling, #9209, 1:1000), beta actin (Cell Signaling, #4967, 1:2000). Bands were visualized using a Gel Doc system (Bio Rad) and quantified with Fiji.

## Generation time assay

Day 1 gravid adults were bleached and eggs were pipetted onto 6 cm NGM plates containing 100 μg/mL Carbenicillin seeded with *E. coli* strain HT115 bacteria. Animals were synchronized by performing an egg lay 3 days prior to the experiment. The egg lay of RAGA-1::AID[Somatic] onto auxin plates was done in the morning due to their developmental delay and the egg lay for all other strains was done in the evening. For the egg lays, 10–20 animals were placed on a control or auxin plate and allowed to lay eggs for 45 minutes and then were removed. Early in the morning 3 days later, 15 L4 animals for each condition were picked onto individual 3.5 cm NG Carb plates that were 1-day seeded with 50 μL of HT115 bacteria. Animals were scored every hour until the first egg was laid.

## Brood size measurement

Animals were bleached onto NGM Carb plates and synchronized by egg lay as mentioned above in "Generation time assay". 15 animals from each condition were singled out onto individual 3.5 cm NG Carb plates seeded with HT115 bacteria at the L4 stage. For the next two days, animals were transferred both in the morning and in the evening to freshly seeded plates. For the following two days, animals were transferred only once. The plates containing the eggs were maintained in the incubator at 20°C for 3 days until the progeny had grown into day 1 adults. The number of progeny on each plate were counted and the counts from each day were summed for each individual parent.

## Body size measurement

Worms were anesthetized on an auxin or control plate without bacteria using 1 mg/mL tetramisole/M9. Once still, worms were imaged on a Zeiss Discovery V8 microscope with Axiocam camera. All animals were imaged in brightfield at 8x with a constant exposure. At least 12 worms were imaged per condition per replicate. Body length was analyzed in ImageJ by drawing a line end-to-end down the midline of the worm and measuring the length of the line. Body

size measurements using the microchamber imaging setup across a gradient of auxin concentrations were performed by the Towbin Lab according to Methods in Stojanovski et al [53].

## Diacetyl attraction assay

The diacetyl chemotaxis assay was performed following the protocol from Margie et al [54]. Three days before the experiment, animals were synchronized by performing an egg lay on either control or auxin plates. On the day of the experiment, fresh 0.2% diacetyl was prepared by diluting diacetyl in ethanol. This 0.2% diacetyl solution was then mixed 1:1 with 0.5M sodium azide for a final diacetyl concentration of 0.1% diacetyl. For the control solution, 100% ethanol was mixed 1:1 with 0.5M sodium azide. On day one of adulthood, animals were washed off the plates with M9 and pelleted by centrifugation at 2,500 rpm for 1 minute. Animals were then washed with M9 three more times. After washing, animals were pipetted onto the center of a 6cm control or auxin plate and excess M9 was quickly and carefully removed with a Kimwipe. Then, the diacetyl and ethanol control solutions were added to opposite corners of the plate. After one hour, the number of animals in the diacetyl and ethanol quadrants were counted. To calculate the chemotaxis index, the number of animals in the ethanol control quadrants was subtracted from the number of animals in the diacetyl quadrants, and this number was then divided by the total number of animals scored. For each experiment, the assay was performed on three plates with 50–150 worms on each.

## Copper aversion starvation assay

The copper aversion starvation protocol was adapted from Campbell et al. [55]. Three days before the experiment, animals were synchronized by performing an egg lay on either control or auxin plates. On the day of the experiment, 2-day HT115 seeded control or auxin plates were prepared with fresh 500mM CuSO4 as described in Campbell et al. To test whether RAGA-1::AID[Neuronal] may have altered food sensing, we added a water barrier control which had a barrier of water in between the animals and the bacterial lawn. On day one of adulthood, animals were transferred to an unseeded control or auxin plate to minimize the transfer of bacteria. The plates were then washed with M9 and animals were pelleted by centrifugation at 2,500 rpm for 1 minute. This was followed by three more washes with M9 before animals were pipetted to the half of the chemotaxis plate without food, or in the case of the no food control, a marked starting side. Each plate had approximately 10 animals. After careful drying with a Kimwipe, the number of worms that crossed the midline barrier was counted every 30 minutes for 4 hours. For each experiment, three plates were done per strain.

## RNA sequencing and differential gene expression analysis

**RNA collection.** Animals were grown on HT115 on NGM Carb plates with 50 worms per plate. On day 1 of adulthood, animals were transferred to control or auxin plates. Animals were transferred to fresh plates on day 2 of adulthood. On day 3 of adulthood, animals were transferred to plates with a thinner bacterial lawn (1-day seeded) and condensed to about 130 animals per plate. Animals were washed off plates using M9 + 0.01% Tween, washed twice using M9 buffer, pelleted and flash-frozen in 600 μL of Qiazol in liquid nitrogen. Four individual biological replicate samples were collected on separate days from separate thaws of worm strains. Samples were stored at -80°C. RNA extractions were performed on 2 biological replicates in parallel at a time. RNA extractions were performed using Qiagen RNeasy Mini Kit and eluted in RNase free $H_2O$. 4 independent biological replicates were sent to the Harvard University Bauer Core Facility for library preparation and RNA sequencing.

**RNA sequencing.** Libraries were prepared using a SciClone G3 NGSx workstation (Perkin Elmer) using the Kapa mRNA HyperPrep kit (Roche Applied Science). Polyadenylated mRNAs were captured using oligo-dT-conjugated magnetic beads (Kapa mRNA HyperPrep kit, Roche Sequencing) from 500 ng of total RNA on a Perkin Elmer SciClone G3 NGSx automated workstation. Poly-adenylated mRNA samples were immediately fragmented to 200-300bp using heat and magnesium. First strand synthesis was completed using random priming followed by second-strand synthesis and A-tailing. A dUTP was incorporated into the second strand to allow strand-specific sequencing of the library. Libraries were enriched and indexed using 9 cycles of amplification (Kapa mRNA HyperPrep kit, Roche Sequencing) with PCR primers, which included dual 8bp index sequences to allow for multiplexing (IDT for Illumina unique dual 8bp indexes). Excess PCR reagents were removed through magnetic bead-based cleanup using Kapa Pure magnetic beads on a SciClone G3 NGSx workstation (Perkin Elmer). The resulting libraries were assessed using a 4200 TapeStation (Agilent Technologies) and quantified by QPCR (Roche Sequencing). Libraries were pooled and sequenced on one Illumina NovaSeq SP flow cell using paired-end, 50 bp reads.

**Differential gene expression analysis.** Raw reads were examined for quality using FastQC(v0.11.5) [56] to ensure library generation and sequencing were suitable for further analysis. To perform additonal quality checks, all reads were aligned to Ensembl assembly of the *C. elegans* (PRJNA13758) genome WBcel235_release104 using STAR (v.2.7.0) [57]. Alignments were checked for biases, evenness of coverage, rRNA content, genomic content of alignments, complexity and other quality checks using a combination of FastQC [56], Qualimap [58], and MultiQC [59]. To quantify the abundance of reads corresponding to each transcript, alignment and quantification was performed using Salmon (1.4.0) [60]. Pairwise comparisons of the differential gene expression between the various conditions was performed using DESeq2 [61] from tximport with a p-vaule cutoff of 0.01. Principal component analysis was performed using DESeq2. WormCat was used for gene set enrichment analysis.

## Pairwise comparison

The following pairwise comparisons were performed: (1) Wild type auxin vs no auxin, (2) Wild type no auxin vs RAGA-1::AID[Neuronal] no auxin, (3) Wild type no auxin vs RAGA-1::AID[Somatic] no auxin, (4) RAGA-1::AID[Neuronal] auxin vs no auxin, and (5) RAGA-1::AID[Somatic] auxin vs no auxin. Pairwise comparisons (1) through (3) mentioned were used to identify genes that changed as a result of auxin treatment or baseline genotype of the strain rather than the result of RAGA-1 degradation. This set of genes that was altered at baseline was then excluded from the DEG list generated by pairwise comparisons (4) and (5).

Normalized counts and differentially expressed genes can be found in **S2 Table**.

All data available at https://www.ncbi.nlm.nih.gov/geo/ under accession number GSE237060: https://www.ncbi.nlm.nih.gov/geo/query/acc.cgi?acc=GSE237060

## Lifespans

Lifespans were conducted at 20˚C on 6 cm NGM plates containing 100 μg/mL carbenicillin and conducted as described in [62]. Briefly, worms were synchronized by timed egg lays using gravid adults and once the progeny reached adulthood, 10 or 20 animals were singled out onto plates for a starting population of 100–120 worms. Worms were transferred to freshly seeded plates at least every other day until day 10 or 11 of adulthood. Survival was scored every other day and worms were scored as dead when they were unresponsive to 3 touches on the head and tail. Worms were censored if they crawled up the wall, bagged, exploded, or got contaminated. Details and statistics can be found in the **S1 Table**.

## RNA interference

*pha-4* and *sfa-1* RNAi constructs came from the Ahringer RNAi library. RNAi experiments were carried out using *E.coli* HT115 bacteria expressing RNAi constructs. Bacteria were grown in LB with 100 μg/ml carbenicillin and 12.5 μg/ml tetracycline overnight. 100 uL of culture was added to NGM plates containing 100 μg/ml carbenicillin 48 hours before use. dsRNA construct expression was induced by adding 100 μL IPTG (100 mM) onto the bacterial lawns at least one hour before placing worms on the plate. Verified sequences of RNAi clones are listed in **S3 Table**. Animals were placed on *pha-4* RNAi at L4 stage. Animals were placed on *sfa-1* RNAi from hatch.

## Statistical analysis

Data were graphed and analyzed in GraphPad Prism 9. For lifespan experiments, survival curves were analyzed using the Log-rank test. For generation time, brood size, body size, RAGA-1::AID::EmGFP levels, and diacetyl attraction assay, outliers were excluded using the ROUT method. A one-way ANOVA followed by a multiple comparisons test was performed on the cleaned data set to identify statistically significant differences between groups. For all figures: * indicates $p < 0.05$, ** indicates $p < 0.01$, *** indicates $p < 0.001$, **** indicates $p < 0.0001$.

## Supporting information

**S1 Fig. Images of midbody region and representative images of partial degradation of RAGA-1::AID.** (A) AID of RAGA-1 in the midbody region. (B) Representative images of the worms with partial degradation of RAGA-1::AID (containing a few remaining puncta) at the corresponding time points. 2.5 hours after auxin treatment, RAGA-1::AID::EmGFP signal is diminished, but 7 out of 21 imaged worms still had between 1 and 20 green puncta. After 6 hours on 0.15 mM auxin, RAGA-1::AID is much more thoroughly degraded with only 6 out of 20 imaged animals still containing 1 or 2 GFP puncta. n = 2, at least 20 worms imaged per condition. White bracket indicates intestinal autofluorescence. Picture in last row shows two worms side-by-side.
(TIF)

**S2 Fig. Phosphorylated RSKS-1 levels following AID of RAGA-1 and LET-363.** (A) Representative western blot of one replicate of P-RSKS-1 levels after AID of RAGA-1. (B) Quantification of all 3 repeats of the western blot indicated in (A). Animals were grown on control or 0.15 mM auxin plates from hatch and harvested at day 1 of adulthood. P-RSKS-1 levels are unaffected by auxin in wild type animals. RAGA-1::AID^Neuronal animals have no significant changes in P-RSKS-1 levels at baseline or upon auxin treatment. P-RSKS-1 is unchanged at baseline in RAGA-1::AID^Somatic animals but is significantly decreased in auxin-treated animals. (C) Representative western blot of one replicate of P-RSKS-1 levels after AID of LET-363. (D) Quantification of all 3 repeats of the western blot indicated in (C). Animals were grown on control plates form hatch, placed on auxin or control plates at day 1 of adulthood, then harvested on day 3 of adulthood. P-RSKS-1 levels are unaffected in wild type animals. In LET-363::AID^Somatic animals, adult-onset auxin-treatment significantly decreases P-RSKS-1 levels. For each replicate, intensity was normalized to the signal in wild type animals. Error bars are plotted at mean ± SEM. * indicates $p < 0.05$, ** indicates $p < 0.01$.
(TIF)

**S3 Fig. Lifespan of RAGA-1 degradation control strains.** (A) Lifespan of RAGA-1::AID and TIR1 strains. Administration of auxin to wild type worms or worms containing AID-tagged

RAGA-1, Somatic::TIR1, or Neuronal::TIR1 alone does not alter their lifespan (p = 0.4515, 0.279, 0.2494 and 0.9632 respectively). n = 3. (B) Maximum lifespan, plotted as the survival of the top 10[th] percentile of the population, is significantly extended for RAGA-1::AID[Somatic] in both the absence and presence of auxin as compared to wild type animals (p < 0.0001 for both). (C) The survival curve for *raga-1(ok386)* mutants sometimes contains a "crash" in which the *raga-1* mutants die even faster than wild type animals early on even though the latter portion of the population is long-lived. We have observed this phenotype in the presence and absence of antibiotics. (TIF)

**S4 Fig. GFP fluorescence of RAGA-1::AID::EmGFP strains with and without TIR1.** (A) Animals were grown on control or 0.15 mM auxin plates from hatch then imaged on day 1 of adulthood. Wild type animals have a baseline level of GFP fluorescence largely originating from intestinal autofluorescence. RAGA-1::AID::EmGFP animals without any TIR1 in the background have a higher GFP intensity than wild type animals, as expected, which is unaffected by auxin treatment. The addition of Neuronal::TIR1 doesn't alter GFP levels. In contrast, the addition of Somatic::TIR1, even in the absence of auxin, significantly decreases GFP fluorescence, indicating auxin-independent degradation of the tagged RAGA-1. The addition of auxin further decreases GFP fluorescence in these animals close to the autofluorescence levels noted in wild type animals. Data points indicate the mean fluorescence for an individual animal. All animals imaged across 3 individual experiments are plotted together in this graph. Error bars are plotted at mean ± SEM. * indicates p < 0.05 and **** indicates p < 0.0001. (TIF)

**S5 Fig. Assessment of sensory behavior after neuronal RAGA-1 degradation.** (A) Diacetyl attraction assay for wild type worms (N2) versus the *osm-6* mutant (PR811) with impaired sensory neuron function. *osm-6* mutants have a significant deficit in the ability to sense diacetyl. n = 3 independent experiments with 3 replicates per experiment. (B) Neuronal AID of RAGA-1 has no effect on *C. elegans*' ability to sense diacetyl (p > 0.05 for all pairwise comparisons). n = 5 independent experiments with 2 to 3 replicate plates per experiment. Data was analyzed with a one-way ANOVA and, for (B), followed by multiple comparisons. (C-E) Copper barrier starvation assay. (C) Animals placed on the side of a plate with no food quickly cross the non-aversive water barrier to get to the side of the plate with food and remain there for the duration of the assay. Neuronal degradation of RAGA-1 doesn't impair this food-sensing phenotype. (D) When placed on a plate with an aversive copper barrier in the middle and a food lawn on the other side, worms will, over time, cross the copper barrier to reach the food. Eventually, almost all animals will have crossed and be located on the food side. *C. elegans* with neuronal loss of RAGA-1 behave the same as wild type worms. (E) When placed on a plate that has no food on either side and an aversive copper barrier in the middle, *C. elegans* will, over time, cross the barrier in search of food so that any given time point some, but not all, of the animals will be over the barrier. Neuronal AID of RAGA-1 has no effect on this searching phenotype. For C-E, the behavior of the strains was analyzed with a two-way ANOVA with repeated measures. No significant differences between any of the strains were found in the graphs shown in C-E with p > 0.05 for all comparisons. (TIF)

**S6 Fig. Body size and generation time for AID control strains.** (A) Body size data for RAGA-1::AID control strains. Auxin treatment doesn't affect body size of RAGA-1::AID or Neuronal::TIR1 animals but Somatic::TIR1 have a mild increase in mean size of 0.045 mm. (B) Generation time for RAGA-1::AID and TIR1 control strains. Auxin treatment doesn't affect the generation time of wild type, RAGA-1::AID, or Neuronal::TIR1 animals. Somatic::TIR1

animals treated with auxin have a mild (3.3 hours on average) delay in development. n = 2 replicates with at least 29 individual measurements for each condition. *** indicates p < 0.001, **** indicates p <0.0001.
(TIF)

**S7 Fig. Growth rate and body volume analysis of larval stages of RAGA-1::AID strain in micro chambers.** Growth of the RAGA-1::AID strain was measured with and without neuronal (*rgef-1p*) and somatic (*eft-3p)* expression of TIR1 in the presence or absence of 1 mM auxin. Individual animals housed in microchambers were imaged to measure (A) time to reach each larval stage, (B) volume at molt, and (C) average growth rate per larval stage. Animals with neuronal RAGA-1 depletion have no impairments in growth measurements throughout the developmental stages. In contrast, somatic depletion of RAGA-1 results in slower progression throughout the larval stages and a slower growth rate. Even in the absence of auxin, the strain expressing TIR1 in the soma has mild growth defects, likely due to auxin-independent degradation of AID-tagged RAGA-1. Data were analyzed by a one-way ANOVA followed by multiple comparisons. All control conditions were analyzed relative to the "No TIR1" group and all conditions on 1mM auxin were analyzed relative to the "No TIR1 + 1 mM auxin" group. Results are not significant (p>0.05) unless otherwise indicated by an asterisk. * indicates p<0.05, ** indicates p <0.01, *** indicates p < 0.001, **** indicates p <0.0001. Each data point represents an individual animal and error bars are plotted at mean ± SEM. Data unavailable for eft-3p::TIR1 strain at hatch and L1 due to technical difficulties.
(TIF)

**S8 Fig. Statistics for LET-363 body size data.** (A-C) These are the same data as in Fig 4C but broken up by day for statistical analysis. Animals were treated with auxin beginning on day 1 of adulthood and body length was measured 24 (A), 48 (B), and 72 (C) hours later. Auxin treatment doesn't affect body size in wild type animals but results in a significant decrease in body size of LET-363::AID^Somatic animals on all days measured. Auxin treatment of LET-363::AID-^Neuronal animals results in a minor (~4%) body size reduction on days 2 (A) and 4 (C) as compared to auxin-treated wild type animals. n = 3 independent replicates with a total of at least 58 individual worms measured for each condition. Each points represents an individual animal (animals from all 3 replicates are combined here) and the black line indicates the mean across all 3 replicates. ns indicates p>0.05, *** indicates p < 0.001, and **** indicates p <0.0001.
(TIF)

**S1 Table. Lifespan Table.**
(XLSX)

**S2 Table. List of differentially expressed genes and normalized counts from RNA sequencing.**
(XLSX)

**S3 Table. Verified sequences of *pha-4* and *sfa-1* RNAi clones.**
(XLSX)

# Acknowledgments

The authors would like to thank the Bauer Core Facility at Harvard University for performing the RNA sequencing. Some strains were provided by the CGC, which is funded by NIH Office of Research Infrastructure Programs (P40 OD010440).

## Author Contributions

**Conceptualization:** Hannah J. Smith, Anne Lanjuin, William B. Mair.

**Formal analysis:** Hannah J. Smith.

**Funding acquisition:** Hannah J. Smith, William B. Mair.

**Investigation:** Hannah J. Smith, Anne Lanjuin, Arpit Sharma, Aditi Prabhakar, Ewelina Nowak, Peter G. Stine, Rohan Sehgal, Klement Stojanovski, Benjamin D. Towbin.

**Methodology:** Hannah J. Smith, Anne Lanjuin, Arpit Sharma, Rohan Sehgal, Klement Stojanovski, Benjamin D. Towbin.

**Project administration:** Hannah J. Smith, Anne Lanjuin, William B. Mair.

**Resources:** Anne Lanjuin.

**Supervision:** William B. Mair.

**Visualization:** Hannah J. Smith.

**Writing – original draft:** Hannah J. Smith, Anne Lanjuin, William B. Mair.

**Writing – review & editing:** Hannah J. Smith, William B. Mair.

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
