## [Decision Letter · Decision Letter 0]

27 Feb 2023

Dear Dr Mair,

Thank you very much for submitting your Research Article entitled 'Neuronal mTORC1 inhibition promotes longevity without suppressing anabolic growth and reproduction in C. elegans' to PLOS Genetics.

The manuscript was fully evaluated at the editorial level and by independent peer reviewers. The reviewers appreciated the attention to an important topic but identified some concerns that we ask you address in a revised manuscript.

We therefore ask you to modify the manuscript according to the review recommendations. Your revisions should address the specific points made by each reviewer.

Yours sincerely,

Stefan Taubert, PhD

Guest Editor

PLOS Genetics

Gregory Copenhaver

Editor-in-Chief

PLOS Genetics

As you can see in the reviews, all 3 reviewers thought the manuscript was of interest and generally well designed and rigorously analyzed. However, several important points must be addressed, either experimentally and/or in writing:

- Consider the possibility (experimentally or at least in discussion) that, rather than tissue specificity, allele strength difference may cause the observed phenotypes, as requested by reviewer #1;

- Reframe the TOR-centric view to a more RAGA-1-centric view as per reviewer #2, to better align with the fact that experiments were focused on the latter;

- Ascertain experimentally whether TOR is actually inhibited in the AID strains (RSKS-1 blots), as requested by reviewer #2;

- Perform some mechanistic followup on at least pha-4 in raga-1::AID, and ideally also in let-363::AID, as requested by reviewer #2;

- Perform additional control experiments, compare with additional relevant strains, and provide information on experimental details on AID strains, promoters used, and auxin treatments, please see reviewer #2's comments for details;

- Provide all data underlying the manuscript, specifically: add a life span data table, as is common practice in the field; this should contain all detailed data on all experimental repeats performed including those presented in the paper in figures as well as replicates shown in supplements or not shown; and add some tables on key gene expression data as supplementary Tables, not just via upload to GEO (this must also be completed).

The reviewers have proposed several additional experiments that would be interesting but are not considered essential, such as investigation of skn-1, of tissue specific gene regulation effects via GFP reporters, and of AMPK's role in this pathway; if such experiments are available this would further strengthen the paper.

Reviewer's Responses to Questions

**Comments to the Authors:**

Reviewer #1: In this manuscript entitled “Neuronal mTORC1 inhibition promotes longevity without suppressing anabolic growth and reproduction in C. elegans”, the authors functionally dissected the effects of neuron-specific inhibition of mTORC1 on the longevity, growth, and reproduction of C. elegans. The authors found that neuron-specific degradation of RAGA-1, an upstream regulator of mTORC1, using AID extended the lifespan of worms. The authors then showed that the neuron-specific depletion of RAGA-1 did not disrupt the growth or reproduction of animals, which was impaired by somatic degradation of RAGA-1. The authors also found that neuronal depletion of LET-363, C. elegans mTOR, promoted the longevity of animals while had no effects on the growth and development of worms. Lastly they analyzed genes whose expression was changed by neuronal and somatic RAGA-1 inhibition using RNA seq, and showed that neuronal depletion of RAGA-1 affects the expression of small number of genes compared with somatic degradation of RAGA-1. Overall, this interesting work demonstrates that specific inhibition of mTOR signaling can promote longevity without defects in growth and reproduction. This study was well designed, the experiments were rigorously done, and the paper was well written. I have only a few major comments, which can further improve the paper.

Major comments

1. One alternative explanation about the key data here is that perhaps neuronal inhibition of mTOR is a proper reduction of function that causes longevity without growth/reproduction defects whereas the somatic inhibition is a strong loss of function that causes severe effects on growth/reproduction and impairs health. In fact, their PCA in Figure 5A indicates typical strong vs. weak allele that affect gene expression differentially. I wonder whether they could test this possibility by using different concentrations of auxin with somatic mTOR signaling depletion strains, and determine whether they could achieve longevity without growth/reproduction defects at some lower concentrations. Or at least they need to add discussion point about this possibility.

2. Related to the comment 1, I wonder whether the authors can analyze their RNA seq data to test whether there are overall qualitative or quantitative differences in gene expression changes by neuronal vs. somatic RAGA-1 depletion. I think there must be both aspects, but the current Figure 5 and description in the text do not address the point.

Minor comments

1. On page 9, please add proper references for auxin-independent degradation of AID-tagged proteins.

2. In figure 3 and figure 4, please add statistical significance among auxin-treated animals. In addition, please doublecheck the panel names of the figure 3 legends. There should be no panel D.

3. In figure 3 – supplement 1, please add proper title for the figure in the legends.

4. On pages 14 and 26, please add a reference for “Stojanovski et al.”

Reviewer #2: In this manuscript, the authors investigate the tissue-specific role of mTORC1 pathway inhibition in extending lifespan. They use the AID degron system to degrade RAGA-1 or LET-363/mTOR in C. elegans neurons only, and find that lifespan extension is achieved without causing other physiological defects associated with whole-body mTOR inhibition such as reduced body size and progeny production. This suggests that lifespan benefits can be uncoupled from the ‘side effects’ of mTOR inhibition. The authors present data using a novel tool for neuron-specific mTOR pathway inhibition, and present a solid manuscript that illustrates the importance of tissue-specific analyses of genes involved in lifespan analyses, but may represent incremental novelty besides the development and validation of new tools.

Major comments:

mTOR inhibition

1. The bulk of the data was produced using the RAGA-1::AID strains, not the LET-363::AID strains. Several changes must be made to the claims in the manuscript to downplay the role of “LET-363/mTOR inhibition” and instead clarify that the findings are a result of RAGA-1 degradation. Unless a full panel of tests is performed in the LET-363::AID strains as was done with the RAGA-1::AID strains, the data in Fig 4 showing the effects of auxin-induced LET-363/mTOR degradation should be used to support the RAGA-1 findings. To this end, the direct role of mTOR inhibition should be downplayed (lines 286-300).

2. Throughout the manuscript, the authors use the term “mTOR inhibition” to refer to AID degradation of either RAGA-1 or LET-363. This term is inappropriate if mTOR inhibition is not actually shown and could be challenging to be shown tissue-specifically, which would represent a significant advance if the authors could show that. Nevertheless, the authors need to test if neuronal mTOR inhibition can be shown by assessing phosphorylation of mTOR substrate RSKS1/S6K by Western blot in control and auxin-treated worms.

Differential expression analyses

1. Fig 5 contains some interesting differential expression data; however, there is no follow-up to these findings. The authors could considereably strengthen their work by showing that some of the upregulated genes are required for lifespan extension. Since pha-4 is required for lifespan extension by let-363 RNAi, and identified as an upregulated upon neuronal RAGA-1 inhibiton, the authors must test whether pha-4 is required for lifespan extension upon neuronal RAGA-1::AID degradation (and ideally let-363 AID). This could also be done with skn-1 and/or its target genes and chromatin remodeling or autophagy genes.

2. In lines 355-359, the authors reference their differential expression data shown in Fig 5, proposing that “neuronal mTORC1 is able to cell nonautonomously regulate gene expression in peripheral tissues (lines 358-359). This claim would be much stronger if at least one example of these cell non-autonomous effects was shown. For instance, gst-4 and/or gcs-1 are targets of SKN-1 expressed in the neurons and muscle, and intestine, respectively. Transcriptional reporters could be used as a readout for cell non-autonomous SKN-1-mediated stress response upon neuronal RAGA-1 degradation.

Neuronal RAGA-1 and LET-363 inhibition

1. The authors show that RAGA-1 is reduced in somatc tissues, but not in neuronal tissues. The authors need to show that RAGA-1 and ideally LET-363 is reduced specifically in neur using their RAGA-1::AIDNeuronal strain. The authors are also two different RAGA-1::AIDneuronal strains, rab-3::TIR1 and rgef-1::TIR1. The authors need to explain the use of two different promoters and compare and contrast the knockdown efficiency. In Fig 1 (and throughout the manuscript), no images are shown of neuronal RAGA-1::AID::EmGFP expression. Images of the head and other body regions of this strain are needed to assess the expression pattern.

Lifespan extension by AID of RAGA-1 in soma and neurons

1. In lines 191-193, authors claim that the lifespan extension seen in RAGA-1::AIDSomatic animals even without auxin treatment may be due to “a low level of RAGA-1 degradation, as mild auxin-independent degradation of AID-tagged proteins has been previously reported”. First, this statement should be referenced. Moreover, this could easily be tested by imaging and comparing fluorescence intensity in stains 1 and 1x3 (as shown in figure 1A) +/- auxin treatment.

2. It is not clear when RAGA-1::AID Somatic animals are placed on auxin in this experiment, This informatin need s to be included in line 194 and the figure legend. Moreover the authors shold include a d1 auxin control into Figure 2A, if this is not the d1 experiment.

3. In Fig 2A, it would be helpful to include raga-1 mutant lifespans +/- auxin as a control. Since there are several raga-1 mutants, as discussed by the authors, perhaps the two with the most and least change in lifespan should be included.

Lifespan data

1. The lifespan table is not included in the manuscript. This is a critical set of data that will have to be assessed, especially since claims throughout the text of the manuscript refer to it.

2. For all lifespan graphs, day of onset of auxin treatment should be clearly indicated in the figure and legend.

Inhibiting mTORC1 in the neurons uncouples aging from anabolic trade-offs.

1. The authors are performing several experiments to show that development, growth, and reproduction are unaffcetec upon neuronal inhibition of RAGA-1. Fig 3 would benefit from raga-1 mutant comparisons as a positive control. In Fig 3 S1, no-auxin controls are missing for the three strains that are shown as treated with ‘auxin from hatch’. These controls are required to show whether the observed differences in development are due to auxin treatment and not construct expression.

2. Figure 3 supplement 2: Suggestion to include RAGA-1::AIDSomatic strain in Fig 3 S2 subfigures, as this is an important comparison to test whether the differences that are statistically significant are meaningful relative to whole-body degradation of RAGA-1. Alternatively, this figure could be deleted, as it is not crucial to the main findings and performed with a different TIR1 expressing strain.

Minor comments:

- Line18 (abstract): change “specifically to C. elegans” to “specifically in C. elegans”

- Line 39-40 (author summary): clarify whether “unwanted side effects such as impaired growth, development and reproduction” applies to multiple model organisms or only C. elegans

- Line 81-83: The athors should define Y allocation theory

- Line 94-95: Define or clarify Drosophila terms used in the introduction and discussion, such as “fat-specific” 9GAL4-UAS”, “fat-body” (line 401-402).

- Line 153: Several degron systems have been developed for use in C. elegans, including mini-AID (mAID) and AID2. The authors should clarify which system was used.

- Line 169: “Fig 3A-D” does not correspond to the actual figures shown. Figure legend also does not match the figure.

- Line 188: change “all somatic tissue” to “all somatic tissues”

- Line 217-218: missing reference for requirement of splicing factor SFA-1.

- Line 257: change “moult” to “molt”.

- Line 259: The Stojanovski et al., reference is not in the reference list.

- Line 276: suggest to change “will be true” to “could be true”.

- Line 297: remove “pro-aging phenotypes” since mTOR-inhibition-induced sickness is not equivalent to aging.

- Line 305: change “72 hours post auxin treatement” to “72 hours post onset of auxin treatment”, since auxin treatment is not stopped.

- Line 371-373: missing reference for “our previous finding … whole body mTORC1 inhibition”.

- Line 397: suggestion to change “sufficient for longevity” to “sufficient to promote longevity”.

- Line 787-788: revise syntax.

- Line 809-812: the significance of the asterisks for statistical comparisons should be stated, as done in line 854.

- Suggestion to combine Fig 2 S2 A and B into one graph to be able to compare RAGA-1::AID conditions to PR811.

- In Fig 2 S2B, statistical comparison between wild type and RAGA-1::AID within treatments is more important than between treatments.

- Suggestion to use linear regression or other function as a statistical model for Fig 2 S2D, as there seems to be some difference between wild type +/- auxin.

- Statistical test used is not indicated in the figure legend for Fig 2 S2A.

- Suggestion to combine Fig 4 C-E into one graph with ‘day of adulthood’ on the x-axis and ‘mean body size +/- error) on the y axis to better represent the differences in trends between somatic LET-363::AID and the other two strains.

- A comparison between the roles of RAGA-1 and LET-363 in the context of the mTOR pathway, lifespan extension and other phenotypes could strengthen the Discussion section. Otherwise, this section could be shortened significantly.

Reviewer #3: This is an interesting and well conducted study by the Mair group together with the Towbin lab. The question that is addressed is an important one and the paper is methodologically sound and appears well conducted. I am not an expert on the specific approach (Auxin-inducible degradation), but the method appears elegant and quite powerful. Using this approach, the authors address the question whether the fitness tradeoffs that are typically associated with lifespan effects (benefits) seen with reduction of mTOR signaling are obligatory. Such tradeoffs are commonly assumed to be mechanistically directly linked to mTOR-related benefits, suggesting that these two effects should be impossible to uncouple. The authors provide strong evidence that this is not in fact the case, at least in C. elegans.

The AUX system requires relatively high concentrations of auxin I was initially somewhat concerned that the observed effects might be impacted by AUX off-target effects. However, the authors have carried out several control experiments and I think the data is quite conclusive. The lifespan effects of neuronal AID RAGA-1 are also quite similar to those previously reported with adult RNAi, again supporting the interpretation.

Minor comments:

The observation that neuronal only degradation of RAGA-1 extends lifespan yet only affects a limited number of target genes (compared to somatic RAGA-1 - as judged by RNAseq) is very interesting. The authors state that data will be made available by upload to GEO – I would suggest (in addition) to provide the DEG statistics in csv or xls format as part of supplementary data.

Previous work has reported that neuronal AMPK is required for lifespan extension by mTOR inhibition in C. elegans. It would interesting to determine if neuronal AMPK is similarly required for lifespan extension by RAGA-1 degradation in neurons using the AUX system.

Given these previous data regarding interactions between neuronal AMPK and mTOR-mediated lifespan determination, I would suggest that this aspect should be at the very least discussed in light of the new results presented here.

**Have all data underlying the figures and results presented in the manuscript been provided?**

Reviewer #1: Yes

Reviewer #2: **No: **No Lifespan table provided to reviewers

Reviewer #3: Yes

PLOS authors have the option to publish the peer review history of their article (what does this mean?). If published, this will include your full peer review and any attached files.

Reviewer #1: **Yes: **SEUNG-JAE V. LEE

Reviewer #2: No

Reviewer #3: No

---

## [Decision Letter · Decision Letter 1]

24 Aug 2023

Dear Dr Mair,

We are pleased to inform you that your manuscript entitled "Neuronal mTORC1 inhibition promotes longevity without suppressing anabolic growth and reproduction in *C. elegans*" has been editorially accepted for publication in PLOS Genetics. Congratulations!

Yours sincerely,

Stefan Taubert, PhD

Guest Editor

PLOS Genetics

Gregory P. Copenhaver

Editor-in-Chief

PLOS Genetics

Comments from the reviewers (if applicable):

Reviewer's Responses to Questions

**Comments to the Authors:**

Reviewer #1: The authors addressed my concerns successfully.

Reviewer #2: Revisions acceptable. Recommend publication.

Reviewer #3: The authors have addressed the few concerns I originally had in full and the manuscript also is further improved compared to the original submission.

**Have all data underlying the figures and results presented in the manuscript been provided?**

Reviewer #1: Yes

Reviewer #2: Yes

Reviewer #3: Yes

PLOS authors have the option to publish the peer review history of their article (what does this mean?). If published, this will include your full peer review and any attached files.

Reviewer #1: **Yes: **Seung-Jae V. Lee

Reviewer #2: No

Reviewer #3: No

**Data Deposition**

http://datadryad.org/submit?journalID=pgenetics&manu=PGENETICS-D-23-00067R1

**Press Queries**

---

## [Editor Report · Acceptance letter]

13 Sep 2023

PGENETICS-D-23-00067R1 

Neuronal mTORC1 inhibition promotes longevity without suppressing anabolic growth and reproduction in *C. elegans*

Dear Dr Mair, 

We are pleased to inform you that your manuscript entitled "Neuronal mTORC1 inhibition promotes longevity without suppressing anabolic growth and reproduction in *C. elegans*" has been formally accepted for publication in PLOS Genetics! Your manuscript is now with our production department and you will be notified of the publication date in due course.

With kind regards,

Jazmin Toth

PLOS Genetics

On behalf of:
